# Large Scale Image Completion via Co-Modulated Generative Adversarial Networks

**Shengyu Zhao**
IIIS, Tsinghua University and Microsoft Research

**Jonathan Cui**
Vacaville Christian Schools

**Yilun Sheng**
IIIS, Tsinghua University and Microsoft Research

**Yue Dong**
IIIS, Tsinghua University

**Xiao Liang**
The High School Affiliated to Renmin University of China

**Eric I-Chao Chang**
Microsoft Research

**Yan Xu**[*]
School of Biological Science and Medical Engineering and Beijing Advanced
Innovation Centre for Biomedical Engineering, Beihang University

## Abstract

Numerous task-specific variants of conditional generative adversarial networks have been developed for image completion. Yet, a serious limitation remains that all existing algorithms tend to fail when handling large-scale missing regions. To overcome this challenge, we propose a generic new approach that bridges the gap between image-conditional and recent modulated unconditional generative architectures via *co-modulation* of both conditional and stochastic style representations. Also, due to the lack of good quantitative metrics for image completion, we propose the new *Paired/Unpaired Inception Discriminative Score (P-IDS/U-IDS)*, which robustly measures the perceptual fidelity of inpainted images compared to real images via linear separability in a feature space. Experiments demonstrate superior performance in terms of both quality and diversity over state-of-the-art methods in free-form image completion and easy generalization to image-to-image translation. Code is available at `https://github.com/zsyzzsoft/co-mod-gan`.

## 1 Introduction

Generative adversarial networks (GANs) have received a great amount of attention in the past few years, during which a fundamental problem emerges from the divergence of development between image-conditional and unconditional GANs. Image-conditional GANs have a wide variety of computer vision applications (Isola et al., 2017). As vanilla U-Net-like generators cannot achieve promising performance especially in free-form image completion (Liu et al., 2018; Yu et al., 2019), a multiplicity of task-specific approaches have been proposed to specialize GAN frameworks, mostly focused on hand-engineered multi-stage architectures, specialized operations, or intermediate structures like edges or contours (Altinel et al., 2018; Ding et al., 2018; Iizuka et al., 2017; Jiang et al., 2019; Lahiri et al., 2020; Li et al., 2020; Liu et al., 2018; 2019a; 2020; Nazeri et al., 2019; Ren et al., 2019; Wang et al., 2018; Xie et al., 2019; Xiong et al., 2019; Yan et al., 2018; Yu et al., 2018; 2019; Yu et al., 2019; Zeng et al., 2019; Zhao et al., 2020a; Zhou et al., 2020). These branches of works have made significant progress in reducing the generated artifacts like color discrepancy and blurriness. However, a serious challenge remains that all existing algorithms tend to fail when handling large-scale missing regions. This is mainly due to their lack of the underlying generative capability — one can never learn to complete a large proportion of an object so long as it does not have the capability of generating a completely new one. We argue that the key to overcoming this challenge is to *bridge the gap between image-conditional and unconditional generative architectures*.

---

[*]Corresponding author

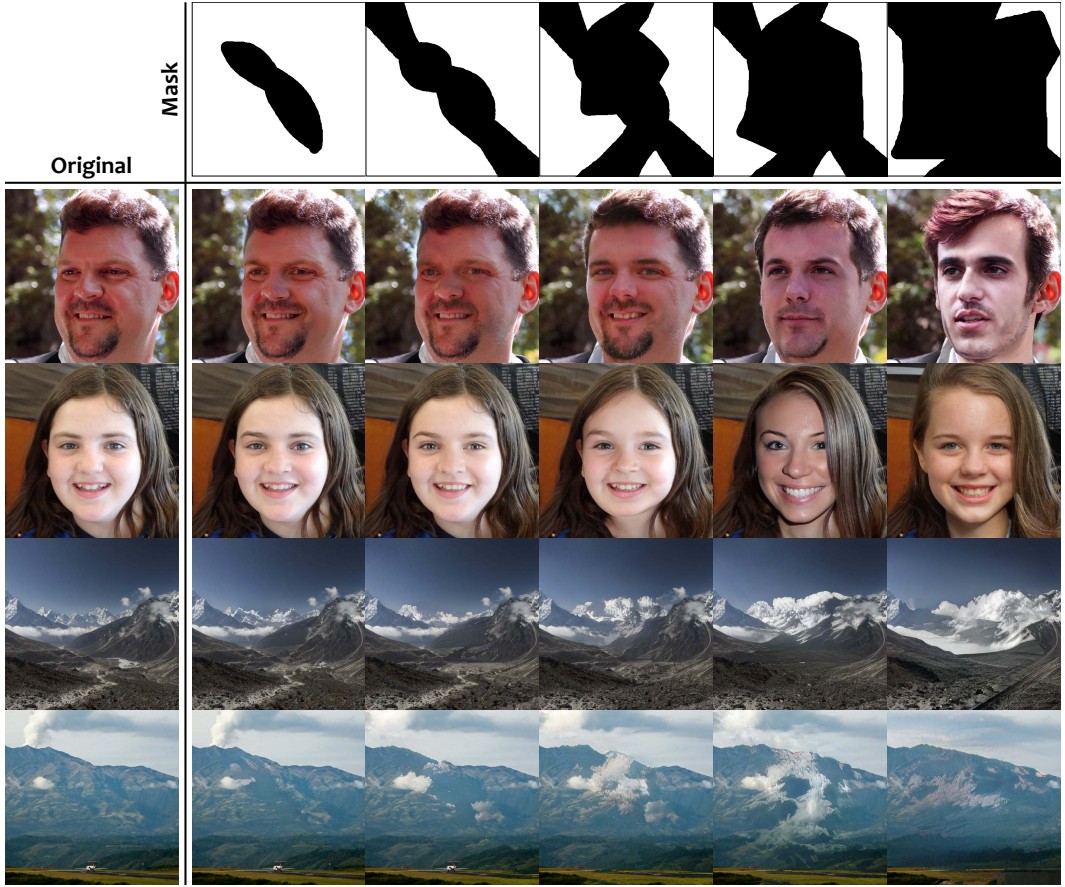

Figure 1: **Our image completion results w.r.t. different masks.** Our method successfully bridges differently conditioned situations, from small-scale inpainting to large-scale completion (left to right). The original images are sampled at $512 \times 512$ resolution from the FFHQ dataset (Karras et al., 2019a) within a 10k validation split (top two examples) and the Places2 validation set (Zhou et al., 2017) (bottom two examples). We refer the readers to the appendix for extensive qualitative examples.

Recently, the performance of unconditional GANs has been fundamentally advanced, chiefly owing to the success of modulation approaches (Chen et al., 2019; Karras et al., 2019a;b) with learned style representations produced by a latent vector. Researchers also extend the application of modulation approaches to image-conditional GANs with the style representations fully determined by an input image (Park et al., 2019; Huang et al., 2018; Liu et al., 2019b); however, the absence of stochasticity makes them hardly generalizable to the settings where only limited conditional information is available. This limitation is fatal especially in large scale image completion. Although some multi-modal unpaired image-to-image translation methods propose to encode the style from another reference image (Huang et al., 2018; Liu et al., 2019b), this unreasonably assumes that the style representations are entirely independent of the conditional input and hence compromises the consistency.

Therefore, we propose *co-modulated generative adversarial networks*, a generic approach that leverages the generative capability from unconditional modulated architectures, embedding both *conditional* and *stochastic* style representations via *co-modulation*. Co-modulated GANs are thus able to generate diverse and consistent contents and generalize well to not only small-scale inpainting but also extremely large-scale image completion, supporting both regular and irregular masks even with only little conditional information available. See Fig. 1 for qualitative examples. Due to the effectiveness of co-modulation, we do not encounter any problem suffered in the image completion literature (Liu et al., 2018; Yu et al., 2019), successfully bridging the long-existing divergence.

Another major barrier in the image completion literature is the lack of good quantitative metrics. The vast majority of works in this literature seek to improve their performance in terms of similarity-based

metrics that heavily prefer blurry results, e.g., $L_1$, $L_2$, PSNR, and SSIM, among which many state that *there are yet no good quantitative metrics for image completion* (Liu et al., 2018; Yu et al., 2018; 2019). The only gold standard in this literature is the user study, which conducts real vs. fake test giving a pair of images to subjects (i.e., *the users*). However, the user study is subject to large variance and costly, therefore lacking reproducibility. Inspired by the user study, we propose the new *Paired/Unpaired Inception Discriminative Score (P-IDS/U-IDS)*. Besides its intuitiveness and scalability, we demonstrate that P-IDS/U-IDS is robust to sampling size and effective of capturing subtle differences and further correlates well with human preferences.

Our contributions are summarized as follows:

- We propose *co-modulated GANs*, a generic approach that bridges the gap between image-conditional and recent modulated unconditional generative architectures.
- We propose the new *P-IDS/U-IDS* for robust assessment of the perceptual fidelity of GANs.
- Experiments demonstrate superior performance in terms of both quality and diversity in free-form image completion and easy generalization to image-to-image translation.

## 2 RELATED WORK

**Image-Conditional GANs.** Image-conditional GANs can be applied to a variety of image-to-image translation tasks (Isola et al., 2017). The unpaired setting is also investigated when paired training data is not available (Choi et al., 2018; Huang et al., 2018; Kim et al., 2019; Lazarow et al., 2017; Liu et al., 2017; Yi et al., 2017; Zhao et al., 2020b; Zhu et al., 2017a). Recent works exploit normalization layers with learned style representations embedded from the conditional input or another reference image to enhance the output fidelity (Huang et al., 2018; Kim et al., 2019; Liu et al., 2019b; Park et al., 2019). They can be regarded as a set of conditional modulation approaches, but still lack stochastic generative capability and hence poorly generalize when limited conditional information is available. Isola et al. (2017) initially find that the generator tends to ignore the noise input although they try to feed it, in contrast to unconditional or class-conditional GANs. A branch of works aims to enforce the intra-conditioning diversity using VAE-based latent sampling strategies (Zhu et al., 2017b) or imposing distance-based loss terms (Huang et al., 2018; Mao et al., 2019; Qin et al., 2018). Wang et al. (2019b) also propose to decompose the convolution kernels into stochastic basis. However, the enforcement of diversity conversely results in the deterioration of image quality. Our co-modulation approach not only learns the stochasticity inherently but also makes the trade-off easily controllable.

**Image Completion.** Image completion, also referred to as image inpainting when incapable of completing large-scale missing regions, has received a significant amount of attention. It is a constrained image-to-image translation problem but exposes more serious challenges. Traditional methods (Ballester et al., 2001; Barnes et al., 2009; Darabi et al., 2012; Efros & Freeman, 2001; Efros & Leung, 1999) utilize only low-level features and fail to generate semantically consistent contents. Then, (Köhler et al., 2014; Ren et al., 2015; Xie et al., 2012) adopt deep neural networks for image completion; (Pathak et al., 2016) first exploits conditional GANs. Numerous follow-up works focus on the semantic context and texture, edges and contours, or hand-engineered architectures (Altinel et al., 2018; Ding et al., 2018; Iizuka et al., 2017; Jiang et al., 2019; Jo & Park, 2019; Lahiri et al., 2017; Liu et al., 2019a; Nazeri et al., 2019; Ren et al., 2019; Sagong et al., 2019; Wang et al., 2018; Xie et al., 2019; Xiong et al., 2019; Yan et al., 2018; Yang et al., 2017; 2019a; Yu et al., 2018; Yu et al., 2019; Zeng et al., 2019; Lahiri et al., 2020; Zhao et al., 2020a; Li et al., 2020; Zhou et al., 2020), among which (Liu et al., 2018; Yu et al., 2019) introduce partial convolution and gated convolution to address free-form image completion. The lack of stochasticity is also observed in image completion (Cai & Wei, 2019; Ma et al., 2019; Zheng et al., 2019). Other works address the so-called outpainting subtasks (Sabini & Rusak, 2018; Wang et al., 2019a; Yang et al., 2019b). To our knowledge, none of these methods produce promising results in the presence of free-form large-scale missing regions.

**Evaluation Metrics.** Great research interest has been drawn on the evaluation of GANs (DeVries et al., 2019; Gurumurthy et al., 2017; Sajjadi et al., 2018; Snell et al., 2017; Xiang & Li, 2017). Inception Score (IS) (Salimans et al., 2016), and some other metrics like FCN-Score (Isola et al., 2017), are specialized to the pre-trained task thus cannot generalize. While FID (Heusel et al., 2017) is generally acceptable, few promising metrics for image completion exist. Previous works heavily rely on similarity-based metrics such as $L_1$, $L_2$, PSNR, and SSIM, which fail to capture stochastic

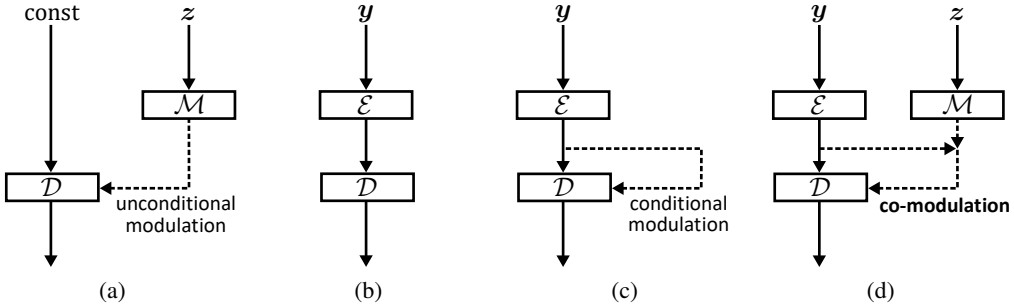

Figure 2: **Illustration from modulation to co-modulation:** (a) unconditional modulated generator; (b) vanilla image-conditional generator; (c) conditional modulated generator; and (d) *co-modulated* generator. $y, z$ represent the conditional input and the latent vector respectively; $\mathcal{E}, \mathcal{D}, \mathcal{M}$ represent the conditional encoder, the generative decoder, and the mapping network, respectively.

regions and are ill-fitted for GANs. Our proposed metric is also related to the classifier-based tests (Blau & Michaeli, 2018; Lopez-Paz & Oquab, 2016). However, previous classifier-based metrics require separate sets for training and testing the classifier, making them sensitive to the underlying generalizability of the trained classifier. We formulate the discriminability as a simple scalable metric for both the paired and unpaired versions without relying on the generalizability.

## 3    CO-MODULATED GENERATIVE ADVERSARIAL NETWORKS

Image-conditional GANs address the problem of translating an image-form conditional input $y$ to an output image $x$ (Isola et al., 2017). We assume for the setting where paired correspondence between input conditions and output images is available in the training data. The generator takes as input an image $y$ along with the latent vector $z$ and produces the output $x$; the discriminator takes as input a pair of $(x, y)$ and seeks to distinguish fake generated pairs from the real distribution. Image completion can be regarded as a constrained image-conditional generation problem where known pixels are restricted to be unchanged. In contrast to the extensive literature on specialized image completion frameworks, we introduce a generic approach that bridges between image-conditional GANs and recent success of unconditional modulated architectures.

### 3.1    REVISITING MODULATION APPROACHES

Modulation approaches emerge from the style transfer literature (Dumoulin et al., 2016; Huang & Belongie, 2017) and are well exploited in state-of-the-art unconditional or class-conditional GANs. They generally apply scalar denormalization factors (e.g., bias and scaling) to the normalized feature maps, while the learned denormalization factors are conditioned on the side information such as class label (Odena et al., 2018) or the latent vector (Chen et al., 2019). Typical normalization layers used in the modulation blocks include batch normalization (Chen et al., 2019; Odena et al., 2018), adaptive instance normalization (Huang & Belongie, 2017; Karras et al., 2019a), and weight demodulation (Karras et al., 2019b) referred to the weight normalization (Salimans & Kingma, 2016).

Here we take StyleGAN2 (Karras et al., 2019b) as an example to show how intermediate activations are modulated as a function of the latent vector. As illustrated in Fig. 2(a), the decoder $\mathcal{D}$ simply originates from a learned constant, while the latent vector $z$ is passed through a multi-layer fully connected mapping network $\mathcal{M}$. The mapped latent vector linearly generates a style vector $s$ for each subsequent modulation via a learned affine transformation $\mathcal{A}$ (i.e., a dense layer without activation):

$$s = \mathcal{A}(\mathcal{M}(z)). \tag{1}$$

Consider a vanilla convolutional layer with kernel weights $w_{ijk}$, where $i, j, k$ enumerate the input channels, the output channels, and the spatial footprint of the convolution, respectively. Given the style vector $s$, the input feature maps are first channel-wise multiplied by $s$, passed through the convolution, and finally channel-wise multiplied by $s'$ where $s'_j = \sqrt{1/\sum_{i,k}(s_i w_{ijk})^2}$ acts as the weight demodulation step that normalizes the feature maps into statistically unit variance.

While modulation approaches have significantly improved the performance of unconditional or class-conditional generators, we wonder whether they could similarly work for image-conditional generators. An intuitive extension to the vanilla image-conditional generator (Fig. 2(b)) would be the conditional modulated generator (see Fig. 2(c)), where the modulation is conditioned on the learned flattened features from the image encoder $\mathcal{E}$. Similar structures also exist in the well-conditioned image-to-image translation tasks (Huang et al., 2018; Liu et al., 2019b; Park et al., 2019). In this case, the style vector can be rewritten as

$$s = \mathcal{A}(\mathcal{E}(\boldsymbol{y})). \tag{2}$$

However, a significant drawback of the conditional modulation approach would be the lack of stochastic generative capability. This problem emerges more apparently in respect of *large scale* image completion. In most cases, the outputs should be *weakly conditioned*, i.e., they are *not* sufficiently determined by the conditional input. As a result, it not only cannot produce diverse outputs but also poorly generalizes to the settings where limited conditional information is available.

### 3.2 CO-MODULATION

To overcome this challenge, we propose *co-modulation*, a generic new approach that easily adapts the generative capability from the unconditional modulated generators to the image-conditional generators. We rewrite the *co-modulated* style vector as (see Fig. 2(d)):

$$s = \mathcal{A}(\mathcal{E}(\boldsymbol{y}), \mathcal{M}(\boldsymbol{z})), \tag{3}$$

i.e., a joint affine transformation conditioning on both style representations. Generally, the style vector can be a non-linear learned mapping from both inputs, but here we simply assume that they can be linearly correlated in the style space and already observe considerable improvements. The linear correlation facilitates *the inherent stochasticity* as will see in §5.1 that co-modulated GANs can easily trade-off between quality and intra-conditioning diversity without imposing any external losses, and moreover, co-modulation contributes to not only stochasticity but also visual quality especially at large-scale missing regions. Co-modulated GANs are encouraged to be trained with regular discriminator losses, while not requiring any direct guidance like the $L_1$ term (Isola et al., 2017), to fully exploit their stochastic generative capability.

## 4 PAIRED/UNPAIRED INCEPTION DISCRIMINATIVE SCORE

Our proposed *Paired/Unpaired Inception Discriminative Score (P-IDS/U-IDS)* aims to reliably measure the linear separability in a pre-trained feature space, inspired by the "human discriminators" in the user study. Let $\mathcal{I}(\cdot)$ be the pre-trained Inception v3 model that maps the input image to the output features of $2048$ dimensions. We sample the same number of real images and their correspondingly generated fake images (drawn from the joint distribution $(\boldsymbol{x}, \boldsymbol{x}') \in \boldsymbol{X}$, where $\boldsymbol{x}$ corresponds to the real image and $\boldsymbol{x}'$ corresponds to the fake image), from which the features are extracted and then fitted by a linear SVM. The linear SVM reflects the linear separability in the feature space and is known to be numerically stable in training. Let $f(\cdot)$ be the (linear) decision function of the SVM, where $f(\mathcal{I}(\boldsymbol{x})) > 0$ if and only if $\boldsymbol{x}$ is considered real. The P-IDS is given by

$$\textbf{P-IDS}(\boldsymbol{X}) = \Pr_{(\boldsymbol{x}, \boldsymbol{x}') \in \boldsymbol{X}} \{f(\mathcal{I}(\boldsymbol{x}')) > f(\mathcal{I}(\boldsymbol{x}))\}, \tag{4}$$

i.e., the probability that a fake sample is considered more realistic than the corresponding real sample.

We also provide an *unpaired* alternative that could generalize to the settings where no paired information is available. We similarly sample the same number of real images (drawn from distribution $X$) and fake images (drawn from distribution $X'$) and fit the linear SVM $f(\cdot)$. We directly calculate the misclassification rate instead:

$$\textbf{U-IDS}(X, X') = \frac{1}{2} \Pr_{\boldsymbol{x} \in X} \{f(\mathcal{I}(\boldsymbol{x})) < 0\} + \frac{1}{2} \Pr_{\boldsymbol{x}' \in X'} \{f(\mathcal{I}(\boldsymbol{x}')) > 0\}. \tag{5}$$

In addition to the super intuitiveness of P-IDS/U-IDS, we would like to emphasize three of their major advantages over FID: the robustness to sampling size, the effectiveness of capturing subtle differences, and the good correlation to human preferences.

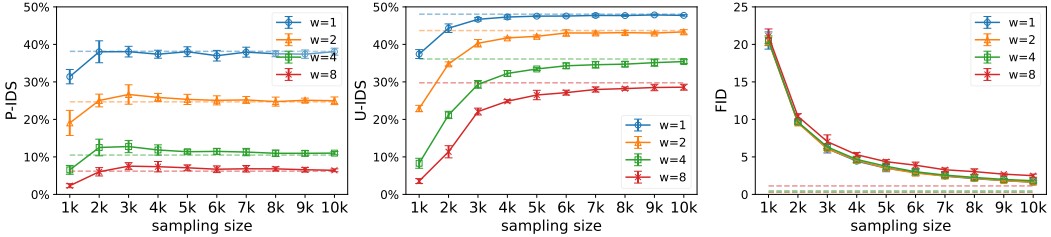

Figure 3: **Robustness to sampling size.** Dashed convergence lines are measured using 50k samples. P-IDS/U-IDS converges fast; FID fails to converge within 10k samples. Results are averaged over 5 runs; error bars indicate standard deviations.

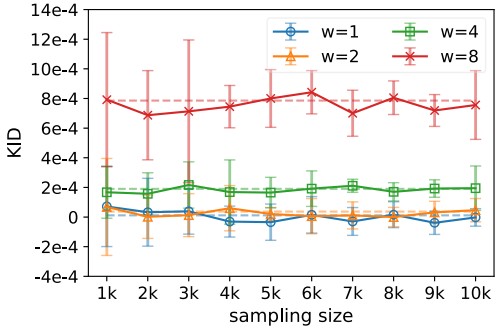 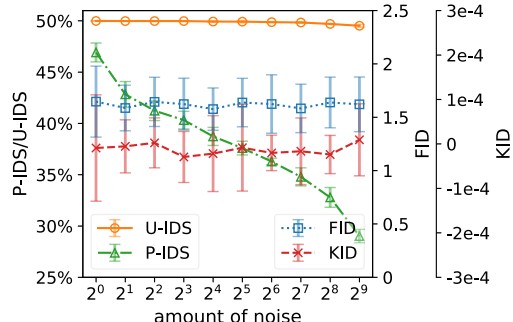

Figure 4: **Kernel Inception Distance (KID) still suffers from large variance.** Although it achieves unbiased estimates, the huge variance even makes them often negative and hardly distinguishable. Results are averaged over 5 runs; error bars indicate standard deviations.

Figure 5: **Effectiveness of capturing subtle differences.** All metrics are measured using 10k samples. P-IDS effectively reflects the amount of noise; FID and KID fail to respond within $2^9$ noisy pixels. Results are averaged over 5 runs; error bars indicate standard deviations.

**Robustness to Sampling Size.** We test the response of P-IDS, U-IDS, FID, and KID to four manipulation strategies: masking the image (to zeros) with a random square of width $w = 1, 2, 4, 8$, respectively. Images are sampled from the FFHQ dataset (Karras et al., 2019a) at $512 \times 512$ resolution. The reference distribution for calculating FID is measured using 50k samples. As plotted in Fig. 3, both P-IDS and U-IDS converge fast within a small number of samples and successfully distinguish the manipulation strategies; FID fails to converge within 10k samples, while the highest convergence line (1.13 when $w = 8$, measured using 50k samples) is even below the lowest FID at 10k samples (1.63 when $w = 1$). Although KID addresses the "biased" problem of FID (Bińkowski et al., 2018), we find that the estimates are still subject to huge variance like FID especially when the two distributions are close. KID requires a fixed block size (Bińkowski et al., 2018) to achieve unbiased estimates; even with a block size of 1000 that minimizes its variance, the estimates are still hardly distinguishable especially between $w = 1$ and $w = 2$ as plotted in Fig. 4.

**Effectiveness of Capturing Subtle Differences.** Capturing subtle differences is particularly important in image completion, since the difference between inpainted and real images only exists in a partial region. We construct subtle image manipulation strategies by masking $n$ random pixels which are then nearest-point interpolated by the neighboring pixels, using the same environment as the last experiment. As plotted in Fig. 5, P-IDS successfully distinguishes the number of manipulated pixels, while FID and KID fail to respond within $2^9$ noisy pixels. We note that U-IDS is still more robust in this case since the central tendency of FID and KID is significantly dominated by the variance.

**Correlation to Human Preferences.** P-IDS imitates the "human discriminators" and is expected to correlate well with human preferences. While it seems clear in Fig. 6, we quantitatively measure the correlation using these data points (20 in total): the correlation coefficient is 0.870 between P-IDS and human preference rate, significantly better than $-0.765$ of FID. Table 3 further provides a case analysis where our P-IDS/U-IDS coincides with clear human preferences as opposed to FID.

Table 1: **Quantitative results for large scale image completion.** Our method is compared against DeepFillv2 (Yu et al., 2019) and RFR (Li et al., 2020). Results are averaged over 5 runs.

| Method | FFHQ | | | Places2 | | |
|---|---|---|---|---|---|---|
| | P-IDS (%) | U-IDS (%) | FID | P-IDS (%) | U-IDS (%) | FID |
| RFR (official) | $0.0 \pm 0.0$ | $0.0 \pm 0.0$ | $48.7 \pm 0.5$ | $0.3 \pm 0.0$ | $4.6 \pm 0.0$ | $49.6 \pm 0.2$ |
| DeepFillv2 (official) | $0.0 \pm 0.0$ | $0.1 \pm 0.0$ | $83.5 \pm 0.6$ | $0.8 \pm 0.0$ | $8.4 \pm 0.0$ | $30.6 \pm 0.2$ |
| DeepFillv2 (retrained) | $0.9 \pm 0.1$ | $8.6 \pm 0.2$ | $17.4 \pm 0.4$ | $1.4 \pm 0.0$ | $11.4 \pm 0.0$ | $22.1 \pm 0.1$ |
| Ours | $\mathbf{16.6 \pm 0.3}$ | $\mathbf{29.4 \pm 0.3}$ | $\mathbf{3.7 \pm 0.0}$ | $\mathbf{13.3 \pm 0.1}$ | $\mathbf{27.4 \pm 0.1}$ | $\mathbf{7.9 \pm 0.0}$ |

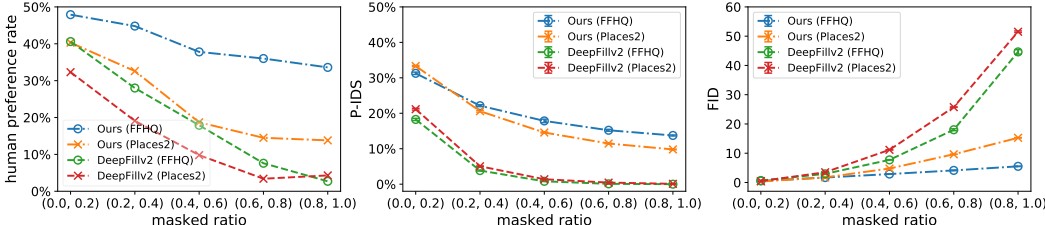

Figure 6: **User study results, P-IDS and FID plots** of DeepFillv2 (retrained) and ours w.r.t. different masked ratios. P-IDS and FID are averaged over 5 runs; error bars indicate standard deviations.

**Computational Analysis.** The time complexity of training a linear SVM is between $O(n^2)$ and $O(n^3)$ (Bottou & Lin, 2007), compared to $O(nd^2 + d^3)$ of FID (Heusel et al., 2017) and $O(n^2d)$ of KID (Bińkowski et al., 2018), where $n$ is the sampling size and $d$ is the dimension of feature space. In practice, P-IDS/U-IDS incurs mild computational overhead in addition to the feature extraction process. For example, with 10k samples, extracting the Inception features on an NVIDIA P100 GPU takes 221s, and fitting the SVM (which only uses CPU) takes an extra of 88s; with 50k samples, the feature extraction process and the SVM take 1080s and 886s respectively.

## 5 EXPERIMENTS

### 5.1 IMAGE COMPLETION

We conduct image completion experiments at $512 \times 512$ resolution on the FFHQ dataset (Karras et al., 2019a) and the Places2 dataset (Zhou et al., 2017). Implementation details are provided in Appendix A. FFHQ is augmented with horizontal flips; Places2 is central cropped or padded. The sampling strategy of free-form masks for training and evaluating is specified in the appendix. We preserve 10k out of 70k images from the FFHQ dataset for validation. Places2 has its own validation set of 36.5k images and a large training set of 8M images. We train our model for 25M images on FFHQ and 50M images on Places2. Our model is compared against RFR (Li et al., 2020) and DeepFillv2 (Yu et al., 2019), the state-of-the-art algorithms for free-form image completion, using both their official pre-trained models and our retrained version of DeepFillv2 (using the official code, our datasets, and our sampling strategy) at 1M iterations (i.e., 32M images). We sample the output once per validation image for all the metrics (P-IDS, U-IDS, and FID). The overall results are summarized in Table 1. Fig. 6 plots the user study results, P-IDS, and FID of DeepFillv2 (retrained) and ours w.r.t. different masked ratios. See Fig. 7 for a qualitative comparison. All these results demonstrate our superior performance. More qualitative examples, numerical user study results and complete tables w.r.t. the masked ratio, and details of the user study are provided in the appendix.

**The Inherent Stochasticity.** Co-modulated GANs are inherently stochastic, i.e., they naturally learn to utilize the stochastic style representations without imposing any external losses, and they are able to produce diverse results even when both the input image and the input mask are fixed. Furthermore, by tuning the truncation $\psi$ (Karras et al., 2019b; Kynkäänniemi et al., 2019) that explicitly amplifies the stochastic branch by $\psi$ times, co-modulated GANs can easily trade-off between quality and diversity (see Fig. 8).

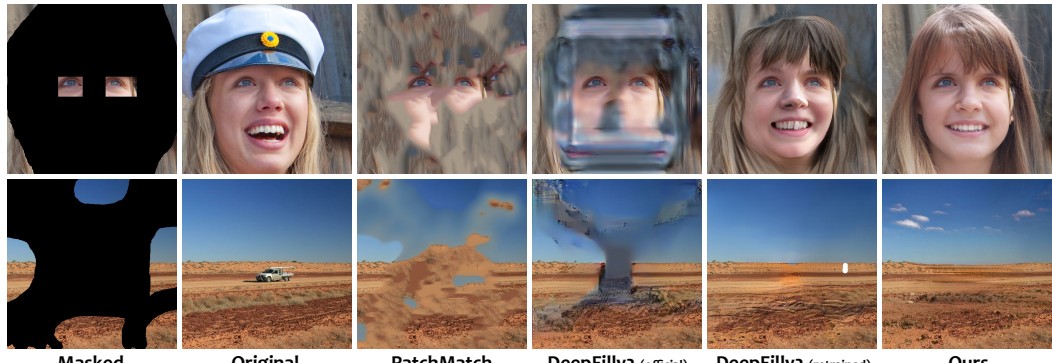

**Masked**  **Original**  **PatchMatch**  **DeepFillv2 (official)**  **DeepFillv2 (retrained)**  **Ours**

Figure 7: **Qualitative examples of state-of-the-art methods on large scale image completion:** PatchMatch (Barnes et al., 2009), DeepFillv2 (Yu et al., 2019), and ours. The original images are sampled at 512×512 resolution from the FFHQ dataset (Karras et al., 2019a) within a 10k validation split (top) and the Places2 validation set (Zhou et al., 2017) (bottom). We refer the readers to the appendix for extensive qualitative examples.

**Ablation Study.** Co-modulation promotes not only stochasticity but also image quality. We compare vanilla, conditional modulated, and co-modulated GANs as illustrated in Figs. 2(b) to 2(d). Experiments are run on the FFHQ dataset with the same setting as § 5.1. While the vanilla version completely fails, our co-modulation approach dominates the conditional modulated version and especially when the masked ratio becomes large (see Fig. 10). We refer the readers to the appendix for the complete results (Table 6). Qualitatively, we often observe some unusual artifacts of the conditional modulated one in the large missing regions (see Fig. 9), which we hypothesize is due to the lack of stochastic generative capability.

## 5.2 IMAGE-TO-IMAGE TRANSLATION

**Edges to Photos.** Co-modulated GANs are generic image-conditional models that can be easily adopted to image-to-image translation tasks. We follow the common setting (DeVries et al., 2019; Wang et al., 2019b) on the edges to photos datasets (Isola et al., 2017) at 256×256 resolution, where FID samples once per validation image (200 in total) and the training set is used as the reference distribution; LPIPS measures the intra-conditioning diversity for which we sample 2k pairs. As summarized in Table 2, our approach easily achieves superior fidelity (FID) over state-of-the-art methods (Huang et al., 2018; Isola et al., 2017; Wang et al., 2019b; Zhu et al., 2017b) despite the fact that MUNIT assumes for the different unpaired setting (Huang et al., 2018), and also superior diversity on the Edges2Handbags dataset by simply tuning the truncation $\psi$ as well as in the trade-off view (see Fig. 11). Our model does not learn to produce diverse outputs on the Edges2Shoes dataset

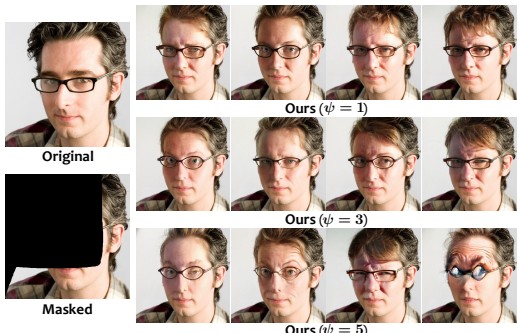

Figure 8: **The inherent stochasticity.** Co-modulated GANs can easily trade-off between quality and diversity by tuning the truncation $\psi$.

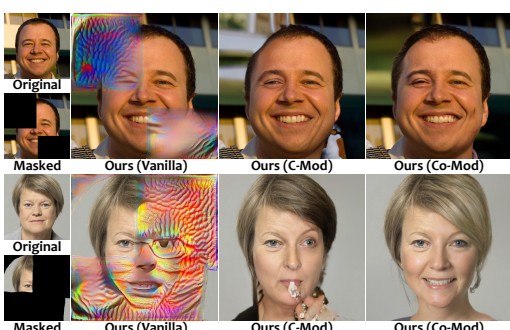

Figure 9: **Qualitative ablation study** among Vanilla, conditional modulation (C-Mod), and our co-modulation (Co-Mod) as in Figs. 2(b) to 2(d).

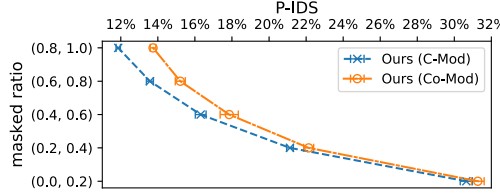 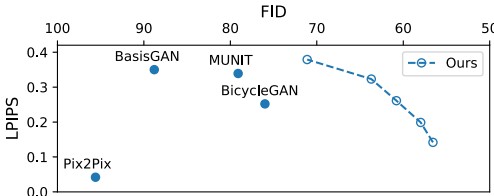

Figure 10: **Co-modulation dominates conditional modulation (C-Mod)** at all masked ratios, especially when the masked ratio becomes large.

Figure 11: **Trade-off curve** of our method between quality (FID) and diversity (LPIPS) on the Edges2Handbags dataset.

Table 2: Image-to-image translation results on the **edges to photos** (Isola et al., 2017) datasets.

| Method | Edges2Shoes | | Edges2Handbags | |
|---|---|---|---|---|
| | FID | LPIPS | FID | LPIPS |
| Pix2Pix (Isola et al., 2017) | 74.2 | 0.040 | 95.6 | 0.042 |
| BicycleGAN (Zhu et al., 2017b) | 47.3 | 0.191 | 76.0 | 0.252 |
| MUNIT (Huang et al., 2018) | 56.2 | 0.229 | 79.1 | 0.339 |
| BasisGAN (Wang et al., 2019b) | 64.2 | **0.242** | 88.8 | 0.350 |
| Ours | **38.5** | 0.036 | **56.9** | 0.143 |
| Ours ($\psi = 3$) | **38.5** | 0.038 | 71.1 | **0.379** |

despite its high fidelity, which we hypothesize is due to the learned strong correspondence between the input edge map and the color information extracted from the limited training set.

**Labels to Photos (COCO-Stuff).**  We further experiment on the COCO-Stuff dataset (Caesar et al., 2018) at 256×256 resolution following the experimental setting of SPADE (Caesar et al., 2018). The real images are resized to a short edge of 256 and then random cropped. The input label map has 182 classes; an embedding layer is used before feeding it into the network. We sample the output once per validation image (5k in total) for all the evaluation metrics. Table 3 shows that our method matches the FID of SPADE but significantly outperforms its P-IDS and U-IDS, without any direct supervision like the perceptual loss used in SPADE. We further conduct a user study between SPADE and ours. The user study indicates consistent human preference of ours over SPADE in accordance with our proposed P-IDS/U-IDS. Qualitative results and the user study details are provided in the appendix.

Table 3: Image-to-image translation results on the **COCO-Stuff** dataset (labels to photos).

| Method | User study (SPADE vs. ours) | P-IDS (%) | U-IDS (%) | FID |
|---|---|---|---|---|
| SPADE (Park et al., 2019) | 41.0% | 1.1 | 5.3 | 22.6 |
| Ours | **59.0%** | **4.5** | **11.3** | **22.5** |

## 6 CONCLUSION

We propose the *co-modulated generative adversarial networks*, a generic approach that bridges the gap between conditional and unconditional modulated generative architectures, significantly improves free-form large scale image completion, and easily generalizes to image-to-image translation. We also propose the intuitive new metric — P-IDS/U-IDS — for robustly assessing the perceptual fidelity for GANs. We expect our approach to be a fundamental solution to the image completion literature and contribute as reliable quantitative benchmarks.

ACKNOWLEDGEMENTS

This work is supported by the National Science and Technology Major Project of the Ministry of Science and Technology in China under Grant 2017YFC0110903, Microsoft Research under the eHealth program, the National Natural Science Foundation in China under Grant 81771910, the

Fundamental Research Funds for the Central Universities of China under Grant SKLSDE-2017ZX-08 from the State Key Laboratory of Software Development Environment in Beihang University in China, the 111 Project in China under Grant B13003.

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

## APPENDIX A  IMPLEMENTATION DETAILS

We mostly borrow the network details and hyperparameters from StyleGAN2 (Karras et al., 2019b), including the number of convolutional layers (2) at each level, the number of channels (64 at 512×512 resolution, doubled at each coarser level with a maximum of 512), architecture of the mapping network $\mathcal{M}$ (8-layer MLP), layer-wise noise injection, style mixing regularization (with a probability of $0.5$ instead), non-saturating logistic loss (Goodfellow et al., 2014) with $R_1$ regularization (Mescheder et al., 2018) of $\gamma = 10$, and the Adam optimizer (Kingma & Ba, 2014) with a learning rate of 0.002.

Our conditional encoder $\mathcal{E}$ imitates a similar architecture as the discriminator but without the cross-level residual connections. Skip residual connections are used between each level of $\mathcal{E}$ and $\mathcal{D}$. To produce the conditional style representation, the final 4×4 feature map of $\mathcal{E}$ is flattened and passed through a fully connected layer of $1024$ channels with a dropout rate of $0.5$. The dropout layer keeps enabled during testing since we observe that it partially correlates to the inherent stochasticity.

Our model has 109M parameters in total. All the experiments are run on 8 cards of NVIDIA Tesla V100 GPUs. The batch size is 4 per GPU, 32 in total. The training length is 25M images unless specified, which takes about 1 week at 512×512 resolution.

## APPENDIX B  FREE-FORM MASK SAMPLING

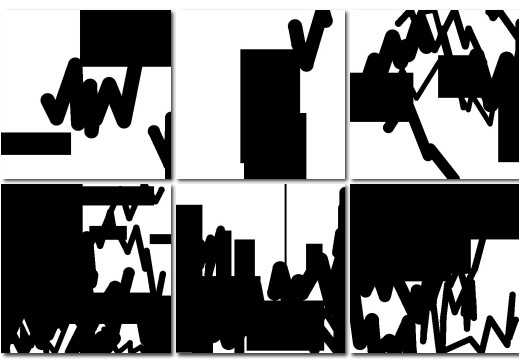

Figure 12: Random samples of free-form masks.

We sample free-form masks for training by simulating random brush strokes and rectangles. The algorithm of generating brush strokes is borrowed from DeepFillv2 (Yu et al., 2019), while the width of the brush is uniformly sampled within $[12, 48]$, the number of vertices is uniformly sampled within $[4, 18]$, and the number of strokes is uniformly sampled within $[0, 20]$. We then generate multiple rectangles with uniformly random widths, heights, and locations, while the number of up to full-size rectangles is uniformly sampled within $[0, 5]$ and the number of up to half-size rectangles is uniformly sampled within $[0, 10]$. See Fig. 12 for the sampled free-form masks. During evaluation, we use the same sampling strategy of free-form masks as used in training if no masked ratio is specified; otherwise, we repeatedly apply the same algorithm until the specified range is satisfied.

## APPENDIX C  USER STUDY

For the user study of image completion, we randomly sample the same number (256) of validation images, free-form masks (using the algorithm above), and the corresponding outputs from each

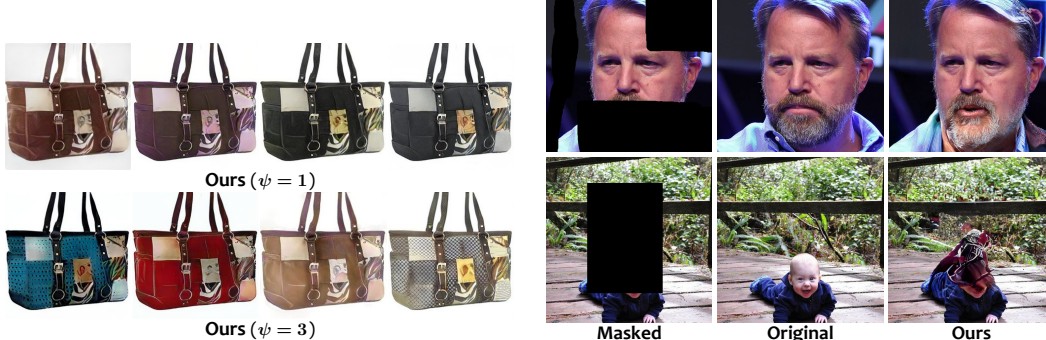

Figure 13: Samples on Edges2Handbags.          Figure 14: Failure cases.

method, for each dataset and each range of masked ratio. The user is given a pair of fake and the corresponding real images in each round and has 5 seconds to decide which one is fake or "don't know"; overtime rounds are also treated as "don't know". No user will see a real for more than once. To compute the user preference rate of fakes over reals, we regard a correct answer as 0, an incorrect answer as 1, and a "don't know" as 0.5. We have received totally 14336 rounds of answers from 28 participants. See Table 4 for the numerical results.

We adopt a similar protocol for the user study on COCO-Stuff. In each round, the user is given a pair of generated images of SPADE (Park et al., 2019) and ours using the same validation input. The user has 5 seconds to decide which one is preferred or "don't know"; overtime rounds are also treated as "don't know". We regard a "don't know" as 0.5. We have received 720 rounds of answers from 12 participants, among which 319 prefer ours, 189 prefer SPADE, and 212 "don't know".

## APPENDIX D   MORE QUANTITATIVE RESULTS

Table 5 presents the quantitative results for image completion across methods and masked ratios. Table 6 presents the quantitative results of the ablation experiment. Experiments demonstrate our superior performance at all masked ratios.

## APPENDIX E   MORE QUALITATIVE RESULTS

Fig. 13 presents our generated samples for image-to-image translation on the Edges2Handbags dataset under both $\psi = 1$ (which achieves superior fidelity) and $\psi = 3$ (which achieves superior diversity). See Fig. 15 for a qualitative comparison for image-to-image translation on the COCO-Stuff dataset. Extensive examples for free-form image completion are presented in Figs. 18-23.

## APPENDIX F   DISCUSSION

Large scale image completion is a challenging task that requires not only generative but also recognition capability. Although our model generates promising results in most of the cases, it sometimes fails to recognize the semantic information in the surrounding areas hence produces strange artifacts (see Fig. 14), especially in the challenging Places2 dataset that contains millions of scenes under various style and quality. The readers are encouraged to discover more examples from our interactive demo.

Table 4: User study results for image completion at different masked ratios among PatchMatch (Barnes et al., 2009), DeepFillv2 (retrained) (Yu et al., 2019), and ours.

| Method | FFHQ | | | | | Places2 | | | | |
|---|---|---|---|---|---|---|---|---|---|---|
| | (0, .2) | (.2, .4) | (.4, .6) | (.6, .8) | (.8, 1) | (0, .2) | (.2, .4) | (.4, .6) | (.6, .8) | (.8, 1) |
| PatchMatch | 10.7% | 2.7% | 3.2% | 1.9% | 2.1% | 15.7% | 4.1% | 3.1% | 1.7% | 3.1% |
| DeepFillv2 (retrained) | 40.6% | 28.0% | 17.9% | 7.6% | 2.7% | 32.3% | 19.1% | 9.8% | 3.4% | 4.3% |
| Ours | **47.9%** | **44.8%** | **37.8%** | **36.0%** | **33.6%** | **40.3%** | **32.7%** | **18.7%** | **14.5%** | **13.8%** |

Table 5: Quantitative comparison for image completion at different masked ratios among DeepFillv2 (official), DeepFillv2 (retrained) (Yu et al., 2019), and ours. Results are averaged over 5 runs.

| Masked Ratio | Method | FFHQ | | | Places2 | | |
|---|---|---|---|---|---|---|---|
| | | P-IDS (%) | U-IDS (%) | FID | P-IDS (%) | U-IDS (%) | FID |
| (0, .2) | DeepFillv2 (official) | $6.4 \pm 0.1$ | $28.8 \pm 0.2$ | $1.78 \pm 0.02$ | $21.3 \pm 0.2$ | $42.6 \pm 0.1$ | $0.51 \pm 0.01$ |
| | DeepFillv2 (retrained) | $18.3 \pm 0.3$ | $40.3 \pm 0.2$ | $0.67 \pm 0.00$ | $21.2 \pm 0.3$ | $42.8 \pm 0.1$ | $0.47 \pm 0.00$ |
| | Ours | $\mathbf{31.3 \pm 0.3}$ | $\mathbf{44.4 \pm 0.3}$ | $\mathbf{0.54 \pm 0.01}$ | $\mathbf{33.3 \pm 0.2}$ | $\mathbf{46.2 \pm 0.0}$ | $\mathbf{0.33 \pm 0.00}$ |
| (.2, .4) | DeepFillv2 (official) | $0.0 \pm 0.0$ | $2.4 \pm 0.1$ | $16.30 \pm 0.11$ | $3.7 \pm 0.1$ | $24.5 \pm 0.1$ | $4.41 \pm 0.03$ |
| | DeepFillv2 (retrained) | $3.8 \pm 0.2$ | $22.0 \pm 0.2$ | $2.95 \pm 0.03$ | $5.1 \pm 0.1$ | $27.1 \pm 0.1$ | $3.50 \pm 0.03$ |
| | Ours | $\mathbf{22.1 \pm 0.3}$ | $\mathbf{37.5 \pm 0.2}$ | $\mathbf{1.69 \pm 0.01}$ | $\mathbf{20.6 \pm 0.3}$ | $\mathbf{38.7 \pm 0.1}$ | $\mathbf{1.74 \pm 0.02}$ |
| (.4, .6) | DeepFillv2 (official) | $0.0 \pm 0.0$ | $0.0 \pm 0.0$ | $46.11 \pm 0.09$ | $0.5 \pm 0.0$ | $10.7 \pm 0.1$ | $14.85 \pm 0.06$ |
| | DeepFillv2 (retrained) | $0.8 \pm 0.0$ | $11.1 \pm 0.1$ | $7.73 \pm 0.09$ | $1.4 \pm 0.0$ | $14.3 \pm 0.1$ | $11.16 \pm 0.05$ |
| | Ours | $\mathbf{17.9 \pm 0.5}$ | $\mathbf{32.1 \pm 0.2}$ | $\mathbf{2.88 \pm 0.03}$ | $\mathbf{14.6 \pm 0.2}$ | $\mathbf{31.2 \pm 0.1}$ | $\mathbf{4.77 \pm 0.03}$ |
| (.6, .8) | DeepFillv2 (official) | $0.0 \pm 0.0$ | $0.0 \pm 0.0$ | $96.58 \pm 0.19$ | $0.1 \pm 0.0$ | $3.1 \pm 0.0$ | $35.81 \pm 0.14$ |
| | DeepFillv2 (retrained) | $0.1 \pm 0.0$ | $3.2 \pm 0.1$ | $18.02 \pm 0.33$ | $0.5 \pm 0.0$ | $6.4 \pm 0.0$ | $25.75 \pm 0.06$ |
| | Ours | $\mathbf{15.2 \pm 0.3}$ | $\mathbf{27.9 \pm 0.2}$ | $\mathbf{4.13 \pm 0.06}$ | $\mathbf{11.5 \pm 0.1}$ | $\mathbf{24.9 \pm 0.0}$ | $\mathbf{9.64 \pm 0.04}$ |
| (.8, 1) | DeepFillv2 (official) | $0.0 \pm 0.0$ | $0.0 \pm 0.0$ | $179.12 \pm 0.40$ | $0.0 \pm 0.0$ | $0.0 \pm 0.0$ | $71.72 \pm 0.16$ |
| | DeepFillv2 (retrained) | $0.0 \pm 0.0$ | $0.0 \pm 0.0$ | $44.63 \pm 0.75$ | $0.1 \pm 0.0$ | $2.1 \pm 0.0$ | $51.52 \pm 0.27$ |
| | Ours | $\mathbf{13.7 \pm 0.2}$ | $\mathbf{24.1 \pm 0.2}$ | $\mathbf{5.52 \pm 0.08}$ | $\mathbf{9.8 \pm 0.1}$ | $\mathbf{19.8 \pm 0.0}$ | $\mathbf{15.27 \pm 0.06}$ |

Table 6: Ablation study among Vanilla, conditional modulation (C-Mod), and our co-modulation (Co-Mod), as illustrated in Figs. 2(b) to 2(d), for image completion on the FFHQ dataset. Vanilla completely fails; our co-modulation dominates the conditional modulated version at all masked ratios and especially when the masked ratio becomes large. Results are averaged over 5 runs.

| Masked Ratio | Method | P-IDS (%) | U-IDS (%) | FID |
|---|---|---|---|---|
| (0.0, 0.2) | Vanilla | $0.6 \pm 0.1$ | $3.0 \pm 0.1$ | $71.72 \pm 0.65$ |
| | C-Mod | $30.7 \pm 0.3$ | $44.3 \pm 0.2$ | $0.55 \pm 0.01$ |
| | Co-Mod | $\mathbf{31.3 \pm 0.3}$ | $\mathbf{44.4 \pm 0.3}$ | $\mathbf{0.54 \pm 0.01}$ |
| (0.2, 0.4) | Vanilla | $0.0 \pm 0.0$ | $0.0 \pm 0.0$ | $152.79 \pm 0.37$ |
| | C-Mod | $21.1 \pm 0.2$ | $36.9 \pm 0.2$ | $1.78 \pm 0.02$ |
| | Co-Mod | $\mathbf{22.1 \pm 0.3}$ | $\mathbf{37.5 \pm 0.2}$ | $\mathbf{1.69 \pm 0.01}$ |
| (0.4, 0.6) | Vanilla | $0.0 \pm 0.0$ | $0.0 \pm 0.0$ | $192.24 \pm 0.44$ |
| | C-Mod | $16.3 \pm 0.3$ | $30.9 \pm 0.3$ | $3.03 \pm 0.02$ |
| | Co-Mod | $\mathbf{17.9 \pm 0.5}$ | $\mathbf{32.1 \pm 0.2}$ | $\mathbf{2.88 \pm 0.03}$ |
| (0.6, 0.8) | Vanilla | $0.0 \pm 0.0$ | $0.0 \pm 0.0$ | $247.68 \pm 1.28$ |
| | C-Mod | $13.6 \pm 0.2$ | $26.2 \pm 0.1$ | $4.42 \pm 0.02$ |
| | Co-Mod | $\mathbf{15.2 \pm 0.3}$ | $\mathbf{27.9 \pm 0.2}$ | $\mathbf{4.13 \pm 0.06}$ |
| (0.8, 1.0) | Vanilla | $0.0 \pm 0.0$ | $0.0 \pm 0.0$ | $286.53 \pm 0.93$ |
| | C-Mod | $11.9 \pm 0.1$ | $22.2 \pm 0.1$ | $5.96 \pm 0.03$ |
| | Co-Mod | $\mathbf{13.7 \pm 0.2}$ | $\mathbf{24.1 \pm 0.2}$ | $\mathbf{5.52 \pm 0.08}$ |

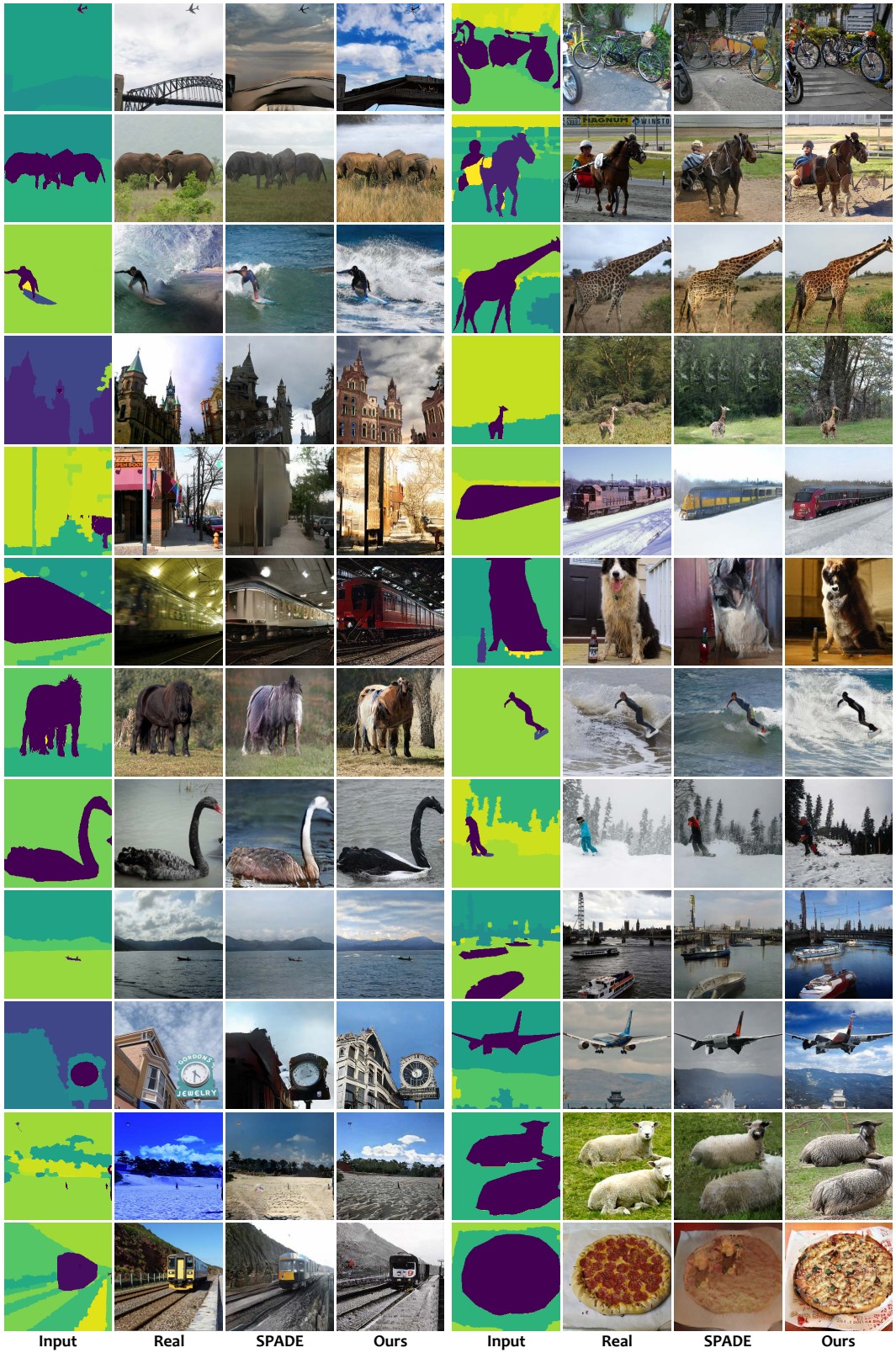

Figure 15: Qualitative comparison between SPADE (Park et al., 2019) and ours for image-to-image translation on the COCO-Stuff validation set (Caesar et al., 2018) (labels to photos).

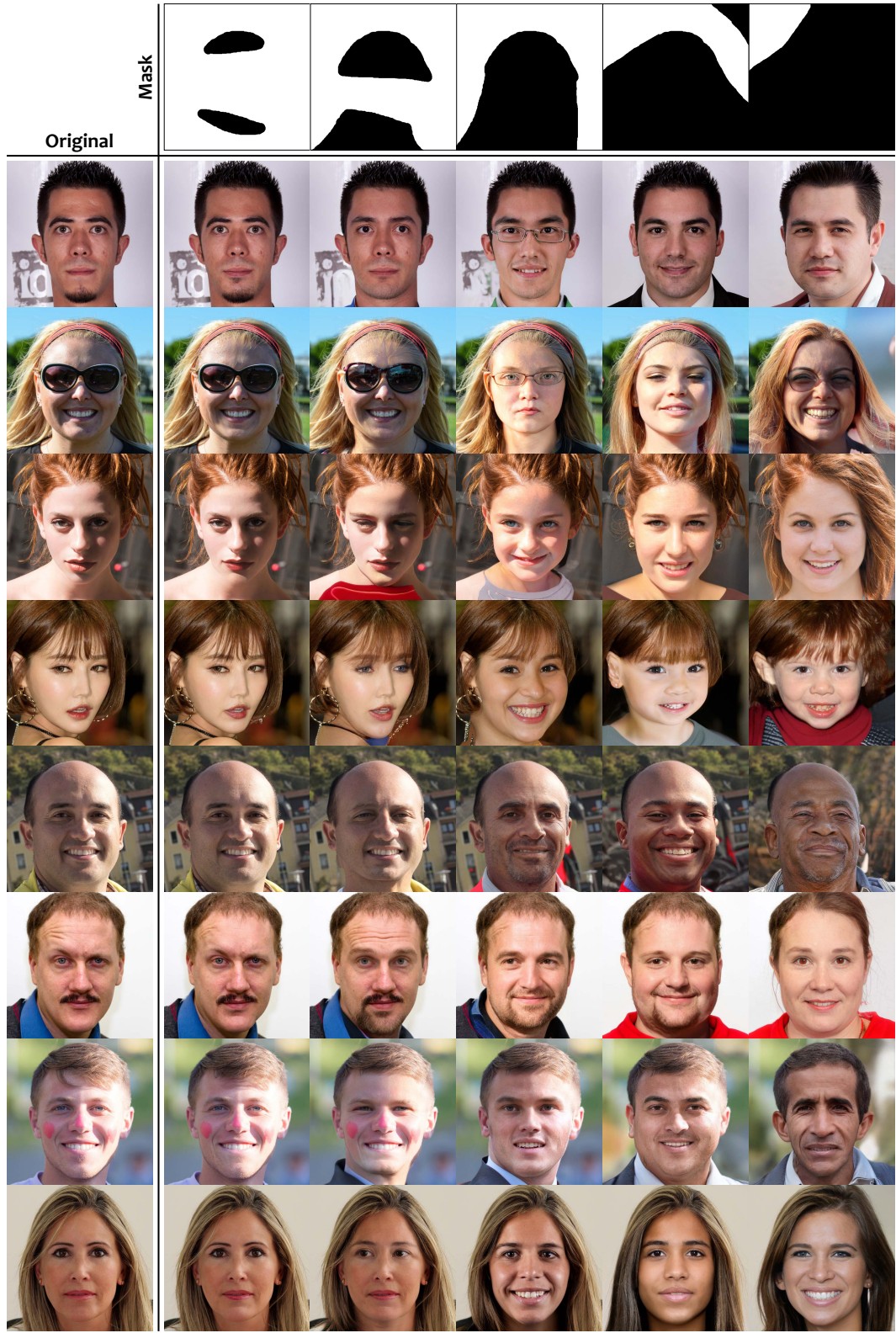

Figure 16: Our image completion results w.r.t. different masks. Our method successfully bridges differently conditioned situations, from small-scale inpainting to large-scale completion (left to right). The original images are sampled at $512 \times 512$ resolution from the FFHQ dataset (Karras et al., 2019a) within a 10k validation split.

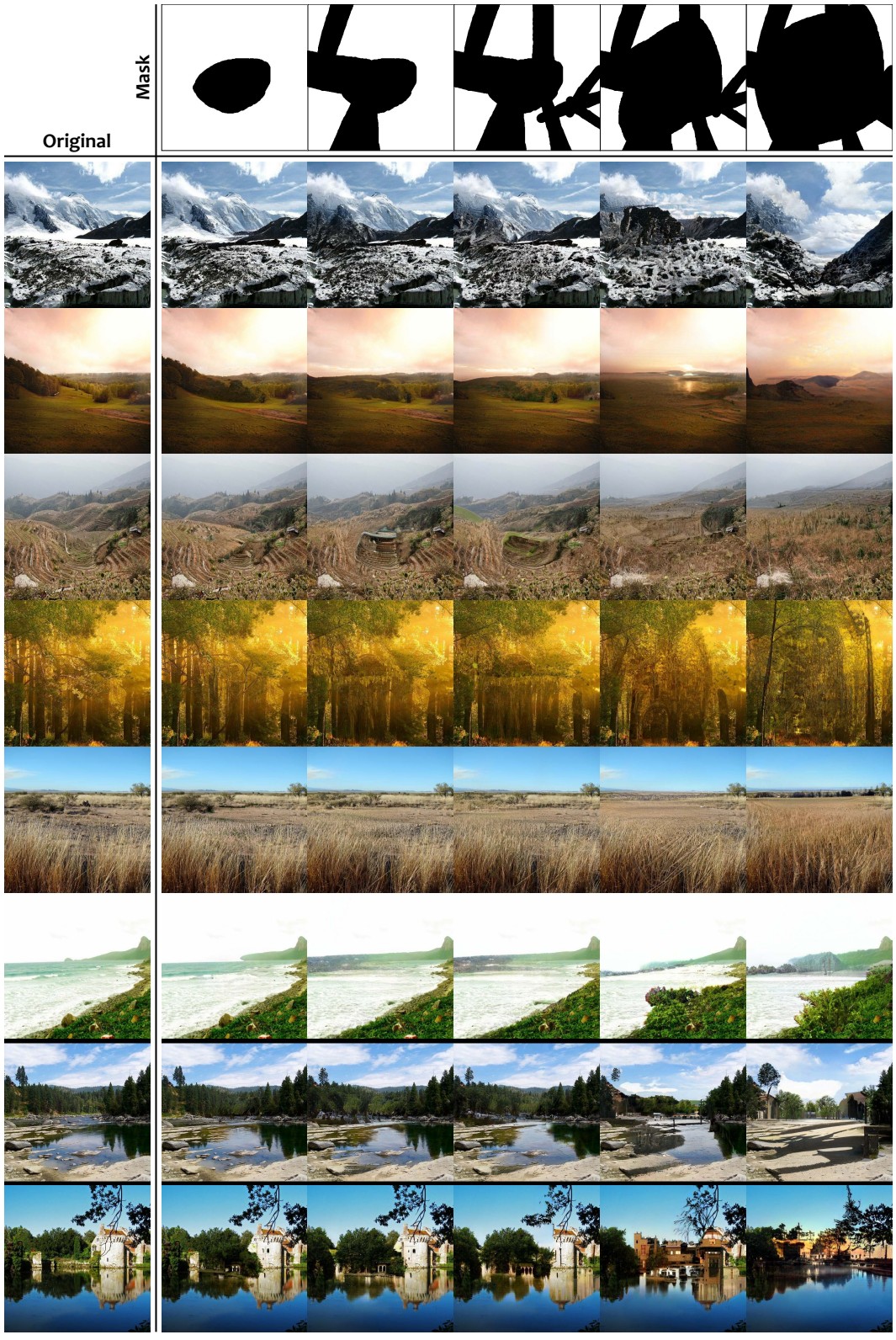

Figure 17: Our image completion results w.r.t. different masks. Our method successfully bridges differently conditioned situations, from small-scale inpainting to large-scale completion (left to right). The original images are sampled at 512×512 resolution from the Places2 validation set (Zhou et al., 2017).

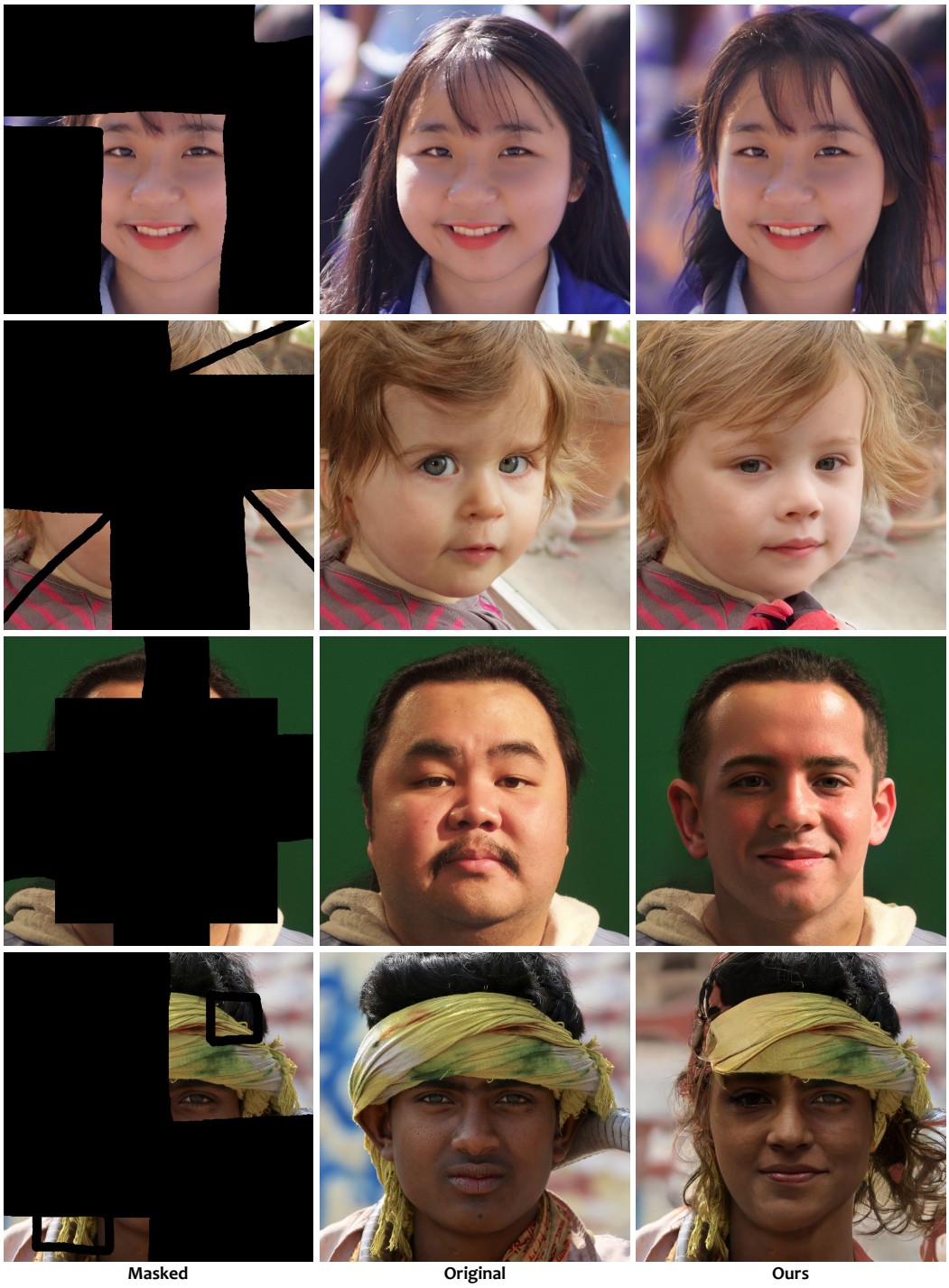

**Masked**         **Original**         **Ours**

Figure 18: Qualitative examples for image completion at 1024×1024 resolution. The original images are sampled from the FFHQ dataset (Karras et al., 2019a) within a 10k validation split.

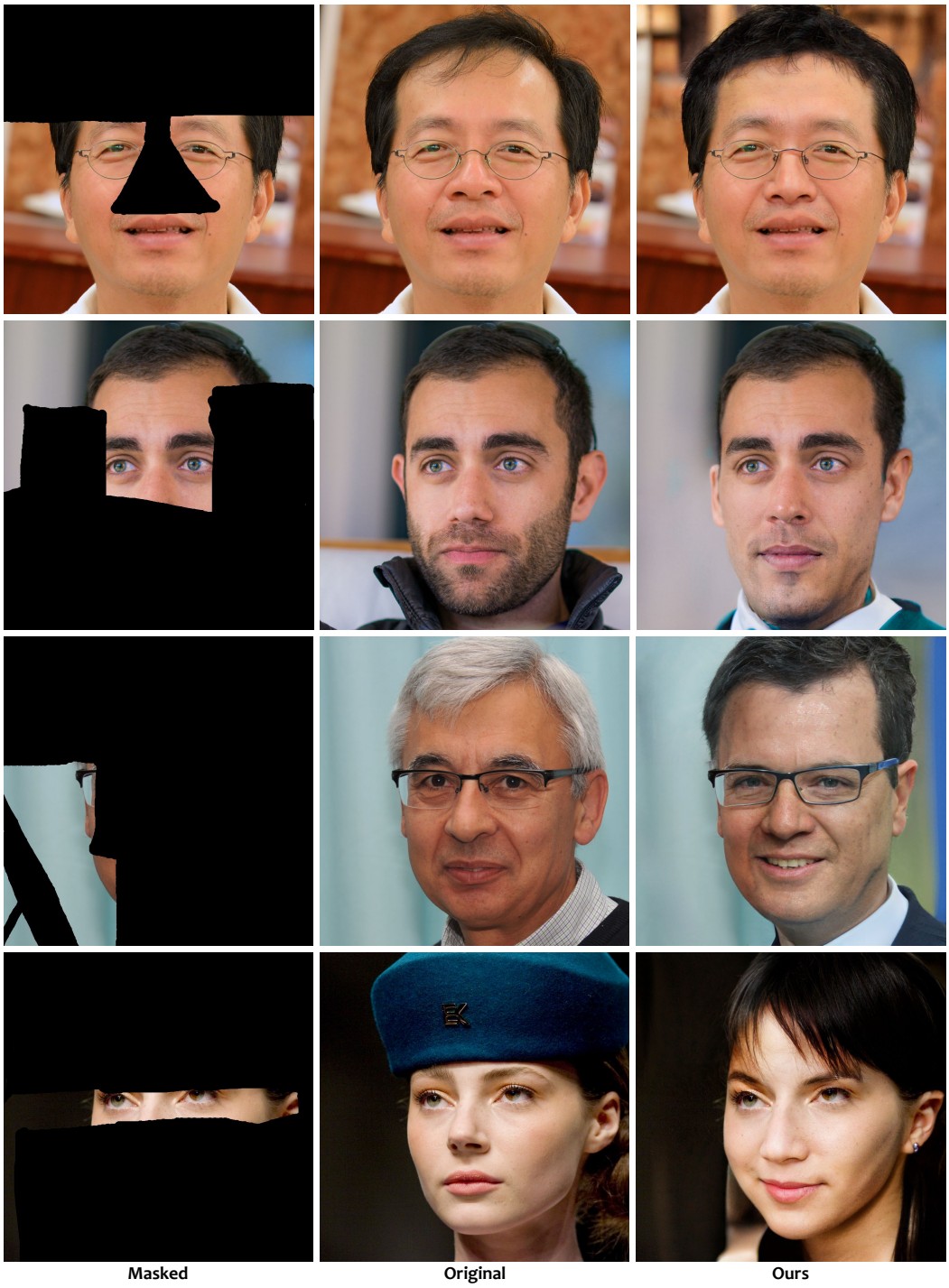

Figure 19: Qualitative examples for image completion at 1024×1024 resolution. The original images are sampled from the FFHQ dataset (Karras et al., 2019a) within a 10k validation split.

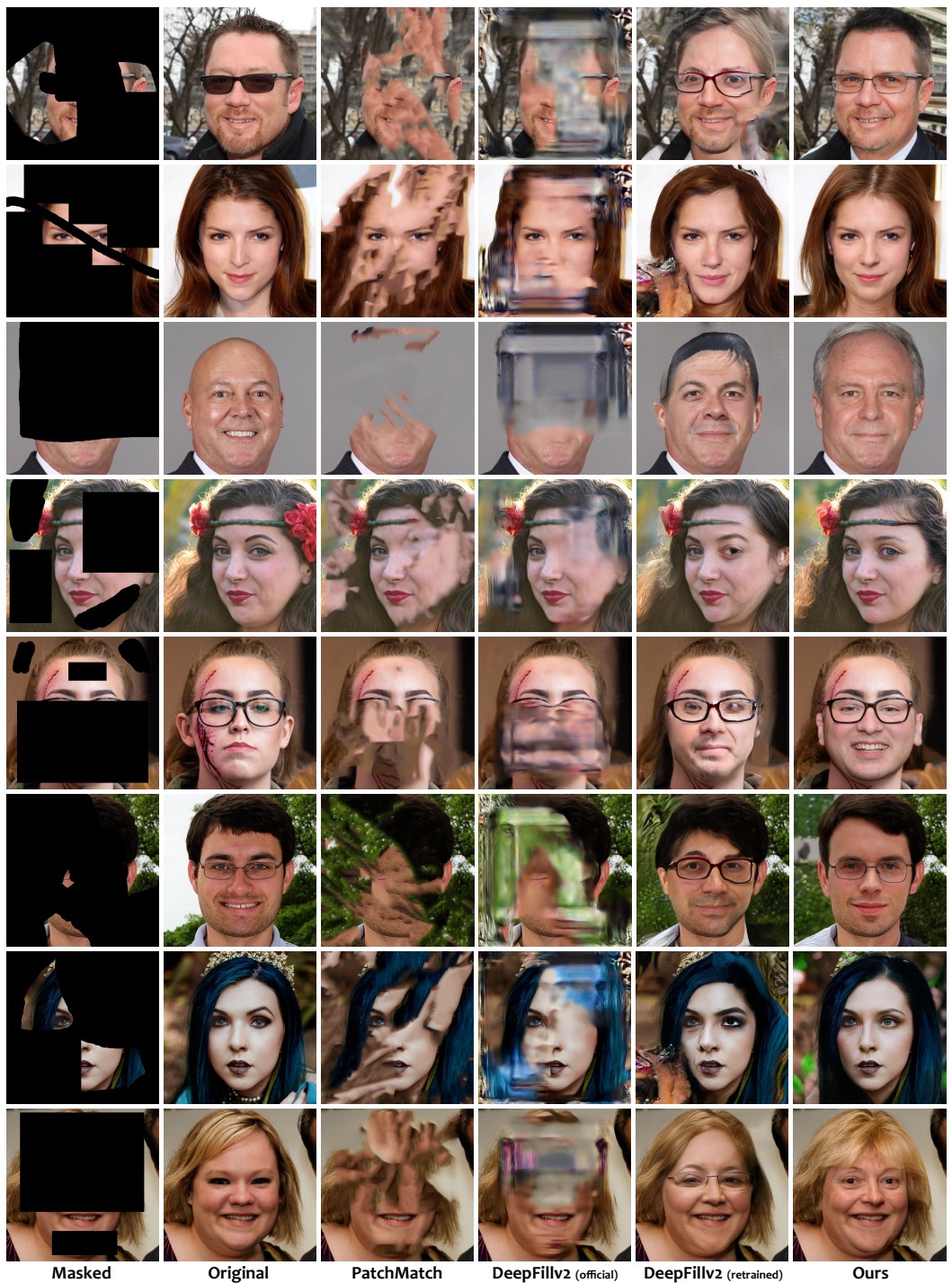

| Masked | Original | PatchMatch | DeepFillv2 (official) | DeepFillv2 (retrained) | Ours |

Figure 20: Qualitative examples for image completion among PatchMatch (Barnes et al., 2009), DeepFillv2 (Yu et al., 2019), and ours. The original images are sampled at 512×512 resolution from the FFHQ dataset (Karras et al., 2019a) within a 10k validation split.

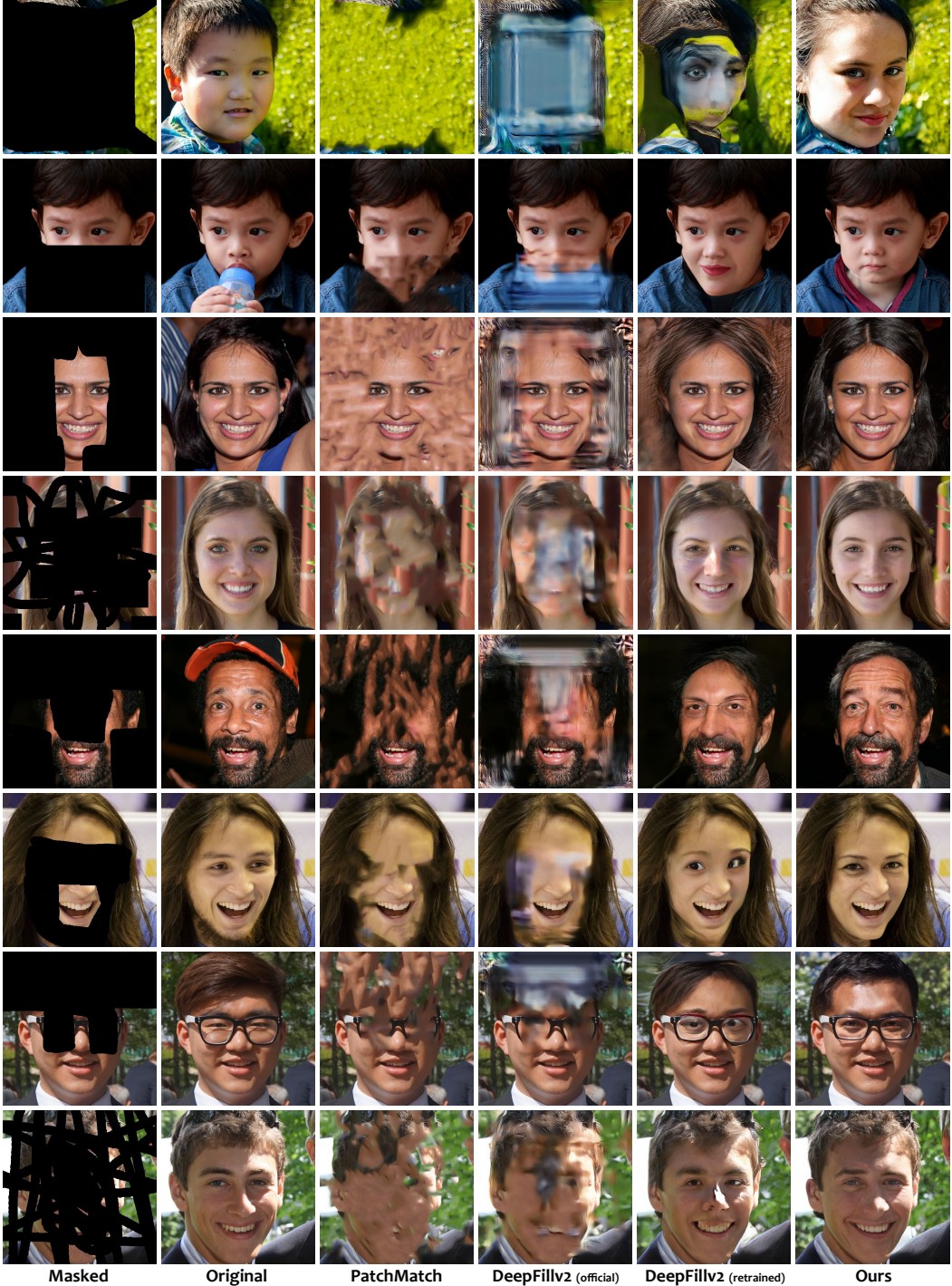

Masked     Original     PatchMatch     DeepFillv2 (official)     DeepFillv2 (retrained)     Ours

Figure 21: Qualitative examples for image completion among PatchMatch (Barnes et al., 2009), DeepFillv2 (Yu et al., 2019), and ours. The original images are sampled at 512×512 resolution from the FFHQ dataset (Karras et al., 2019a) within a 10k validation split.

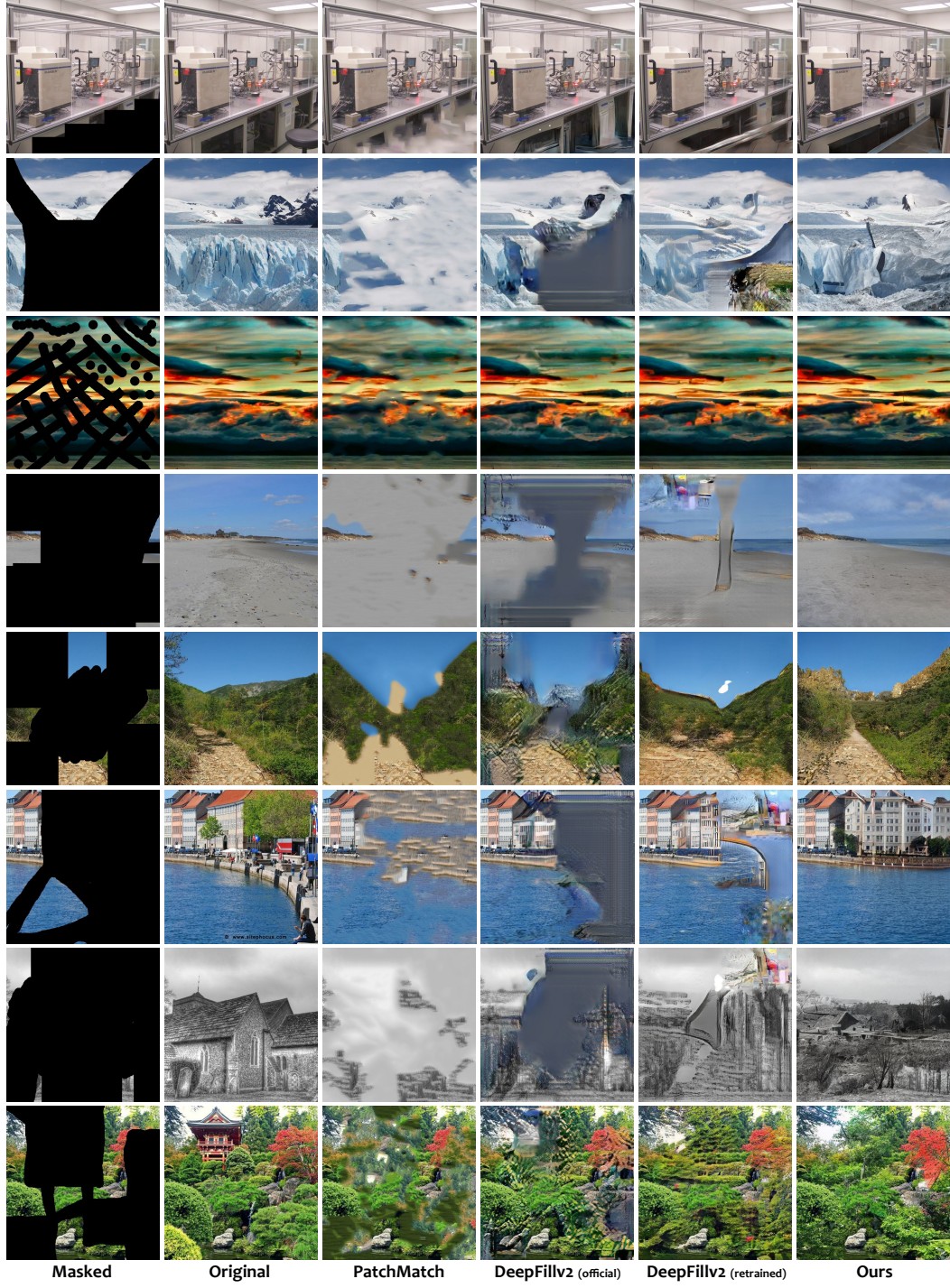

Figure 22: Qualitative examples for image completion among PatchMatch (Barnes et al., 2009), DeepFillv2 (Yu et al., 2019), and ours. The original images are sampled at 512×512 resolution from the Places2 validation set (Zhou et al., 2017).

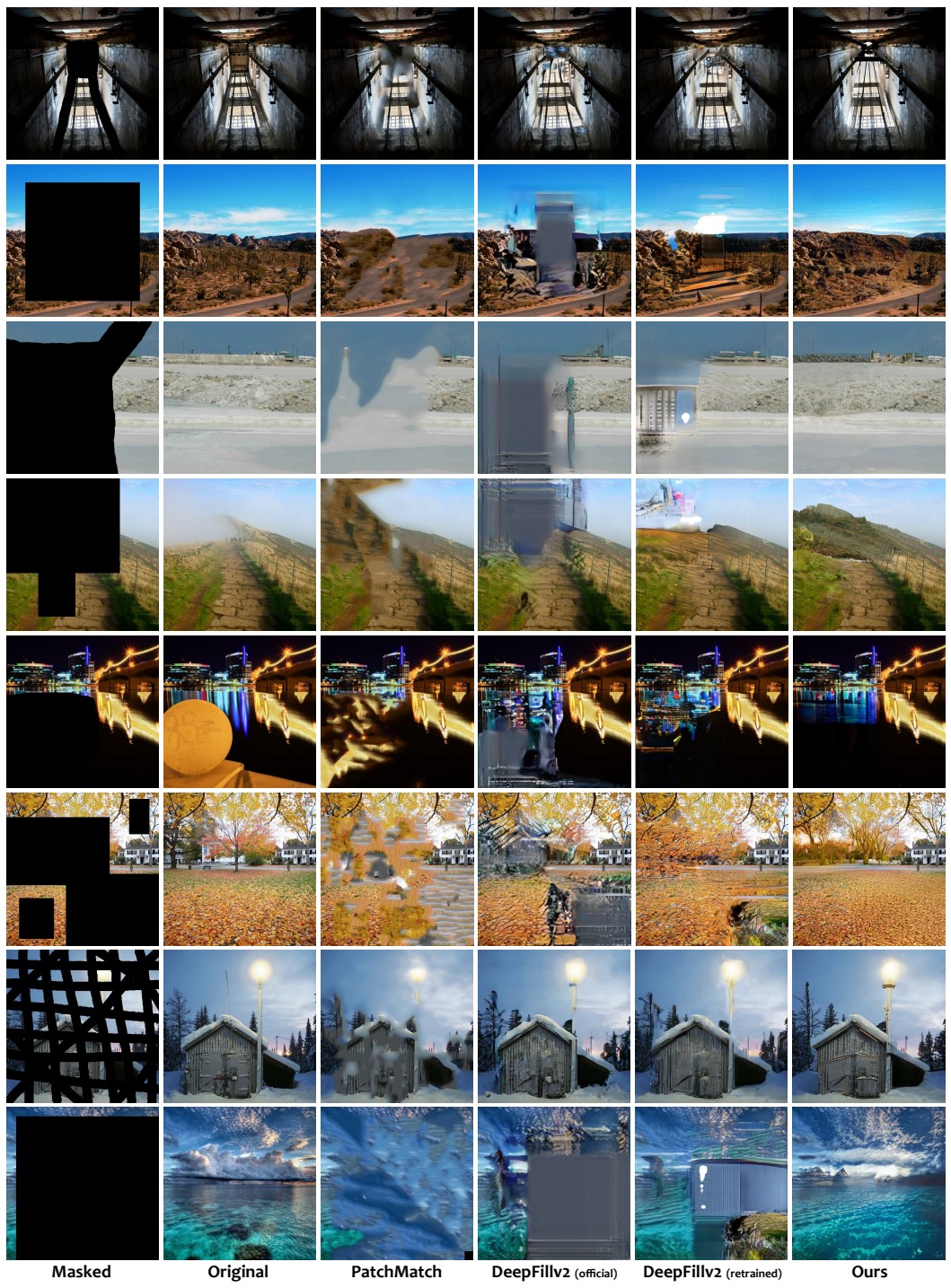

Figure 23: Qualitative examples for image completion among PatchMatch (Barnes et al., 2009), DeepFillv2 (Yu et al., 2019), and ours. The original images are sampled at $512 \times 512$ resolution from the Places2 validation set (Zhou et al., 2017).

