# OpenReview forum: "Large Scale Image Completion via Co-Modulated Generative Adversarial Networks"
_ICLR.cc/2021/Conference — ICLR 2021 Spotlight_

### Official Review · AnonReviewer1 · 2020-10-27
**Review round 0 of “ Large Scale Image Completion via Co-Modulated Generative Adversarial Networks”**

**Rating:** 7
**Confidence:** 3

**Review:**

In this article, authors proposes to bridge the gap between image-conditional and recent modulated unconditional generative architectures with a generic co-modulated gan architecture.
They also proposes a new inception score based on linear SVM in order to measure linear separability in a pre-trained feature space.

The paper is easy to read and the experimentation part supports the claims made in the theoritical sections.

I have one question though: Why using a SVM in the computation of Paired/Unpaired Inception Discriminative Score (P-IDS/U-IDS) and not a simpler linear model ?
How the training of the SVM goes especially on large datasets ? (not in term of accuracy but in term of computation power)

---

> ### Author Response · Authors · 2020-11-17
> **Response to R1**
>
> Thank you very much for your constructive comments.
>
> **1. SVM vs. linear model**
>
> We choose linear SVM because it is a simple type of linear model that is numerically stable and has a standard implementation, hence being more scalable.
>
> **2. Computational cost of SVM**
>
> The time complexity of training a linear SVM is between $O(n^2)$ and $O(n^3)$, where $n$ is the sampling size. In practice, SVM incurs mild computational overhead compared to the feature extraction process. For example, with 10k samples, extracting the Inception features on an NVIDIA P100 GPU takes 221s, and fitting the SVM (which only uses CPU) takes an extra of 88s. With 50k samples, the feature extraction process and the SVM take 1080s and 886s respectively.
>
> We have also listed our major changes in our general response above. Please don’t hesitate to let us know for any additional comments on the paper.

---

### Official Review · AnonReviewer3 · 2020-10-29
**A simple yet effective idea with great results**

**Rating:** 8
**Confidence:** 3

**Review:**

This paper proposes an image completion method that can deal with large-scale missing regions. The proposed method employs a co-modulation technique to bridge the gap between conditional and unconditional GANs. It can then take the advantages from both sides with the consistency offered by conditional GANs and stochasticity provided by the unconditional GANs. The paper also proposes new image quality metrics, Paired/Unpaired Inception Discriminative Scores (P-IDS/U-IDS), for measuring the image quality of the inpainted images. Experiments show that the proposed method significantly outperforms DeepFillv2, a state-of-the-art freeform image completion method, on examples with large missing regions. Overall, I like the idea and the paper’s results. It represents a clear progress to the image completion problem with large missing regions.

**Strengths**
+ The idea is novel and makes sense. It is simple yet effective to the target problem. The idea of extending modulation to co-modulation is interesting. The co-modulation architecture design allows the proposed method to explore the generation capability of the unconditional GANs while maintaining the consistency of the completed image.
+ The results are excellent. The paper and the appendix's visual results show that the proposed method can inpaint images with large missing regions very well.
+ The paper is generally well-written. Figure 2 illustrates the basic idea very well.

**Weaknesses**
- Although the experiments show a good correlation of the proposed P-IDS/U-IDS metrics to the user study results, it is not clear how well the metrics reflect perceptual fidelity. Do these metrics work best for measuring the quality of inpainted images? How were the fake images generated for fitting the linear SVM? Is it possible to generalize to other image restoration problems?
- Although the main target is image completion, the paper claims that the proposed co-modulated GANs also works well on image-to-image translation. However, the paper only demonstrates the application on a simple edge2image problem, from edge images to commodity images. It would be better if the paper presents more image-to-image translation examples for validating how well the proposed method can handle image-to-image translation.

**Post-rebuttal**

The rebuttal addresses the raised issues. An experiment supports the generalizability of the proposed metric. A new image-to-image translation experiment on COCO-Stuff is added. It shows the proposed method's potential for other translation tasks (the results are in Fig. 16, not Fig. 15). Taking the rebuttal and other reviews into account, I would like to recommend accepting the paper for its excellent results and simple yet effective idea.

---

> ### Author Response · Authors · 2020-11-17
> **Response to R3**
>
> Thank you very much for your encouraging and constructive comments.
>
> **1. Generalizability of the proposed P-IDS/U-IDS**
>
> We have tested P-IDS/U-IDS in a new image-to-image translation experiment on COCO-Stuff in the revision (see Table 3). This experiment shows that our metric is generalizable to image-to-image translation and correlates well to human preferences even more than FID. Beyond image-to-image translation, we believe that U-IDS is also a good alternative to FID for assessing unconditional GANs. For example, on FFHQ, we test that the U-IDS of StyleGAN is 17.4%, and the U-IDS of StyleGAN2 is 27.8%. This could be potentially more intuitive than their difference in FID.
>
> **2. How were the fake images generated for fitting the linear SVM?**
>
> In Figs. 4-6, we use the manipulated real images with the corresponding manipulation strategies as detailed in Section 4 as the fake images for fitting the linear SVM. In the image completion experiments, the fake images are the inpainted images generated by the model.
>
> **2. More image-to-image translation experiments**
>
> We have conducted a new image-to-image translation experiment on COCO-Stuff in the revision (see Table 3 and Fig. 15). The user study, P-IDS, U-IDS, and FID consistently validate our superiority over SPADE. Meanwhile, our method does not require any task-specific design or direct supervision like the perceptual loss used in SPADE.
>
> We have also listed our major changes in our general response above. Please don’t hesitate to let us know for any additional comments on the paper.

---

### Official Review · AnonReviewer4 · 2020-10-29
**Literature research is questionable**

**Rating:** 4
**Confidence:** 3

**Review:**

The paper proposes multiple contributions.

The paper identifies the following problem: current inpainting methods are suitable for small missing regions, but do not do well for large missing regions. I think the exposition is outdated and does not consider new work published at CVPR 2020, ECCV 2020, and possibly other venues.
The paper "Recurrent Feature Reasoning for Image Inpainting" explicitely points out "However, filling in large continuous holes remains difficult due to the lack of constraints for the hole center." and the results seem to show many examples of inpaiting for large missing regions.
"Image2StyleGAN++ ..." show the inpainting of large regions as application and specifically mention the large-scale inpainting problem.
"Image Fine-grained Inpainting" states that "Benefited from the property of this network, we can more easily recover large regions in an incomplete image"
"DeepGIN: Deep Generative Inpainting Network for Extreme Image Inpainting" even mentions the problem of large missing regions in the title of the paper.
There are also other inpainting papers that one should look at. I didn't check the publication date or the relationship in detail at this point in time: "Rethinking Image Inpainting via a Mutual Encoder-Decoder with Feature Equalizations", "Deep Generative Model for Image Inpainting with Local Binary Pattern Learning and Spatial Attention", "Deep Generative Model for Image Inpainting with Local Binary Pattern Learning and Spatial Attention", "Encoding in Style: a StyleGAN Encoder for Image-to-Image Translation".
Main concern for this submission is the literature research and I would like to see this addressed in the rebuttal.

The paper proposes an architecture modification the authors call co-modulation. The idea is to have the normalization of the generator layers not only controlled by either a random vector, or an input image, but by both. I think this overall idea is clear, but the details (including Fig. 3) are not that clear. I would say it's a nice, but smaller idea, that is suitable for publication if it leads to good results.

The paper also proposes a new way to evaluate GANs using the proposed P-IDS and U-IDS score. The main idea here is to use a pre-trained feature transformation (the inception network) on real and fake images and then to evaluate the images using a linear classifier.
I do not think the evaluation strategy of masking a random square of size wxw is that convincing. A square of 1x1 is a single pixel and this is not a meaningful image manipulation. Even changing a square of 8x8 pixels is not especially meaningful. The fact that your metric can distinguish between these manipulations is not an indication that the metric can distinguish between high quality and low quality results.
The user study is more meaningful and gives some indication that the new metric is better. This looks promising and I like this result.

---

> ### Author Response · Authors · 2020-11-17
> **Response to R4**
>
> Thank you very much for your constructive comments.
>
> **1. Literature research**
>
> We agree that the problem that current methods are incapable of inpainting large-scale missing regions has been noticed by multiple works, but this problem still remains unsolved. Qualitatively, all those methods mentioned by the reviewer [1-7] either require sparsely masked regions to achieve reasonable results, or generate significant artifacts even when a single medium-scale region is being masked, no matter how they state their statements. Quantitatively, we evaluate the official released models of RFR [1] but get poor results (see our Table 1 in the revision). Since they have not released the Places2 model, we use their Paris StreetView model for Places2 evaluation and their CelebA model for FFHQ evaluation. This comparison is not totally fair but does have some implications.
>
> **References:**
>
> [1] Li et al. "Recurrent Feature Reasoning for Image Inpainting." CVPR 2020.
>
> [2] Abdal et al. "Image2StyleGAN++: How to Edit the Embedded Images?" CVPR 2020.
>
> [3] Hui et al. "Image fine-grained inpainting." arXiv preprint.
>
> [4] Li et al. "DeepGIN: Deep Generative Inpainting Network for Extreme Image Inpainting." arXiv preprint.
>
> [5] Liu et al. "Rethinking Image Inpainting via a Mutual Encoder-Decoder with Feature Equalizations." ECCV 2020.
>
> [6] Wu et al. "Deep Generative Model for Image Inpainting with Local Binary Pattern Learning and Spatial Attention." arXiv preprint.
>
> [7] Richardson et al. "Encoding in Style: a StyleGAN Encoder for Image-to-Image Translation." arXiv preprint.
>
> **2. Details of co-modulation**
>
> Details of the conditional encoder $\mathcal{E}$ and the mapping network $\mathcal{M}$ are provided in Appendix A. The affine transformation $\mathcal{A}$ is implemented as a dense layer without activation. Then the style vector $\mathbf{s}$ (in Eq. (3)) is used to modulate each convolutional layer in the generative decoder as described in Section 3.1. We have made clarifications in the revision.
>
> **3. Analysis of the proposed P-IDS/U-IDS**
>
> We conducted distortion analysis (including masking and noise) to show some properties of our proposed metric: the robustness to sampling size and the effectiveness of capturing subtle differences. Although they may not be that meaningful in practice (for which we rely on the correlation to user study), hopefully they could reveal some interesting properties.
>
> We have also listed our major changes in our general response above. Please don’t hesitate to let us know for any additional comments on the paper.

---

### Official Review · AnonReviewer2 · 2020-10-29
**Review for Large Scale Image Completion via Co-Modulated Generative Adversarial Networks**

**Rating:** 8
**Confidence:** 5

**Review:**

In this paper, the authors propose a general approach for image completion with large-scale missing regions. The key is to combine image-conditional and modulated unconditional generative architectures via co-modulation. The presented approach has demonstrated strong performance in the image painting with large-scale missing pixels and some image-to-image translation tasks. A new metric P-IDS/U-IDS is proposed to evaluate the perceptual fidelity of inpainted images.

Strength:
- The idea of co-modulation is quite interesting and has demonstrated strong results in various tasks.
- The paper is well-written and well-motivated.
- The solution combines the best of two worlds in image-conditional and unconditional image generation.
- A new metric P-IDS/U-IDS is proposed for perceptual evaluation, verified by the correlation to human preferences.

Weakness:
- Only one image inpainting DeepFillv2 is compared in the experiment. Other image inpainting methods such as gated convolution, partial convolution can be also evaluated.
- The results in the main paper are mainly faces that have structured information. It would be good to move some outdoor results in the supplement to the main paper.

---

> ### Author Response · Authors · 2020-11-17
> **Response to R2**
>
> Thank you very much for your encouraging and constructive comments.
>
> **1. Comparison with other inpainting methods**
>
> The DeepFillv2 baseline uses GatedConv, and its experiment has demonstrated superiority to prior works e.g. PartialConv. We further compare with a newer method [1] using their official models. We have added this result in Table 1 in the revision.
>
> **2. Visualization of outdoor results in the main paper**
>
> Thanks for the suggestion. We have added a new figure (Fig. 1) in the revision.
>
> We have also listed our major changes in our general response above. Please don’t hesitate to let us know for any additional comments on the paper.
>
> **Reference:**
>
> [1] Li et al. "Recurrent Feature Reasoning for Image Inpainting." CVPR 2020.

---

### Official Review · AnonReviewer5 · 2020-11-08
**Good performance, novelty is a bit limited.**

**Rating:** 6
**Confidence:** 4

**Review:**

Update: Thanks for the response from the authors. The comments 3/4/5 from authors convincingly address my concerns. Regarding other classifier-based metrics, note that not all of them requires separate training and testing procedure. For example, the leave-one-out 1-NN accuracy [1] does not. Also, I'm still not sure about the technical novelty. Therefore, I keep my original rating.

[1] Xu, Qiantong, et al. "An empirical study on evaluation metrics of generative adversarial networks." arXiv preprint arXiv:1806.07755 (2018).

---------

This paper proposes generator architecture for image inpainting using co-modulation, which is similar to the weight modulation in StyleGAN2 but is conditioned on both the input image and the stochastic variable instead of only the stochastic variable.


Pros:
+ The paper is well-written and easy to read.
+ The proposed method significantly outperforms the baseline DeepFillv2. The results look qualitatively convincing.

Cons:
- The novelty of this paper is a bit limited. It seems to me that the proposed co-modulation is a straightforward extension of conditional modulation with stochasticity.
- It seems most of the improvement over DeepFillv2 is from the better and larger network backbone (StyleGAN2). The improvement from conditional modulation is a bit small (Table 5 in the appendix).
- In Figure 5, the authors argue that KID is subject to huge variance. However, it is not clear to me if KID really has a larger variance than FID. Figure 6 shows that FID and KID have very similar (relative) variance. It is argued that KID can hardly distinguish w=1 vs w=2. But from Figure 4 right, it seems to me that FID has the same problem too.

Questions for the rebuttal:
- What is the running time of the proposed classification-based metrics?
- The proposed co-modulation follows the “late-fusion” strategy: processing the image and random information independently, then concatenate them before modulation. Is it helpful to fuse them earlier, using an MLP to process the concatenated information before modulation?
- The paper argues that "however, to the best of our knowledge, we are the ﬁrst to formulate the discriminability as ψ a simple scalable metric" when discussing other classifier-based metrics. I'm not totally convinced why the proposed metric is more scalable than other ones, e.g., the two-sample test. It would be great if you could elaborate on that.

Overall, I find the empirical results of the paper are pretty impressive. The technical novelty is however a bit lacking.

---

> ### Author Response · Authors · 2020-11-17
> **Response to R5**
>
> Thank you very much for your constructive comments.
>
>
> **1. Novelty**
>
> Despite the simplicity and intuitiveness of co-modulation, we believe that this is an important step to bridge the stochastic generative capability from unconditional to image-conditional GANs.
>
>
> **2. Ablation study of co-modulation**
>
> How to utilize a better unconditional GAN backbone (e.g., StyleGAN2) for solving image completion has been non-trivial and is the focus of this work. Our ablation study (now Table 6 in the appendix) indicates that the Vanilla version (i.e., without modulation) of the StyleGAN2 backbone completely fails. Although the conditional modulation (C-Mod) variant also achieves reasonable performance, even this strategy has not been realized in the image completion literature, while our co-modulation approach (Co-Mod) dominates its performance especially when the masked ratio becomes large.
>
>
> **3. Variance of KID and FID**
>
> We agree that FID also has the problem of large relative variance before convergence. We have clarified this in the revision.
>
>
> **4. Running time of the proposed metric**
>
> P-IDS/U-IDS incurs mild computational overhead in addition to the feature extraction process. For example, with 10k samples, extracting the Inception features on an NVIDIA P100 GPU takes 221s, and fitting the SVM (which only uses CPU) takes an extra of 88s. With 50k samples, the feature extraction process and the SVM take 1080s and 886s respectively. We have included this in the revision.
>
>
> **5. Is it helpful to fuse the conditional and stochastic information earlier?**
>
> We tried to feed the conditional information $\mathcal{E}(\mathbf{y})$ into the mapping network together with the latent vector $\mathbf{z}$ but did not observe clear improvement. On the other hand, our “late-fusion” strategy enables easy trade-off between quality and diversity by linearly magnifying the stochastic term right before modulation; there would not be an easy way to do this if we fuse them earlier when they are highly entangled.
>
>
> **6. Discussion about other classifier-based metrics**
>
> Previous classifier-based metrics (e.g., C2ST-NN/KNN used in the two-sample test) require separate sets for training and testing the classifier. This makes them sensitive to the underlying generalizability of the trained classifier, e.g., whether a model may receive a good score depends not only on the fidelity of its generated images but also on the degree to which the classifier overfits the sampled training set. In comparison, we restrict the classifier to be a simple linear model, which shares with distance-based metrics (like FID and KID) the same property that a given set of features can be measured without relying on the generalizability. We are also the first to formulate the paired discriminability as a scalable metric (P-IDS). We have clarified this in the revision.
>
> We have also listed our major changes in our general response above. Please don’t hesitate to let us know for any additional comments on the paper.

---

### Author Response · Authors · 2020-11-18
**Our General Response**

We thank all the reviewers for their constructive comments and general encouragement. R4 is concerned that the problem that current methods are incapable of inpainting large-scale missing regions has also been identified by recent works. We agree that this is a generic problem in the image completion literature and has been noticed recently, but this problem has never been solved.

Below, we list our major changes in the revision:

**1. Literature research**
- We cited more papers when discussing the problem that current methods are incapable of inpainting large-scale missing regions.

**2. Experiments**
- We conducted a new image-to-image translation experiment on COCO-Stuff in comparison with SPADE [1] in Table 3.
- We added a new image inpainting baseline RFR [2] in Table 1.
- We analyzed the computational complexity and the running time of our proposed metric in the “Computational Analysis” paragraph in Section 4.

**3. Visualization**
- We added a new figure for better visualization in the main paper (Figure 1).
- We visualized the new image-to-image translation experiment on COCO-Stuff in Figure 15 in the appendix.

**References:**

[1] Park et al. "Semantic image synthesis with spatially-adaptive normalization." CVPR 2019.

[2] Li et al. "Recurrent Feature Reasoning for Image Inpainting." CVPR 2020.

---

### Decision · Program_Chairs · 2021-01-07
**Final Decision**

**Decision:**

Accept (Spotlight)

**Comment:**

This paper received two clear accept, one accept, one borderline accept and one reject review. R4 identified that the paper falls short in discussing recent works from CVPR and ECCV 2020 on the image inpainting and completion tasks which also tackle challenging scenarios in these tasks. The authors improve their related work section with these more recent works while pointing out that the task still remains unsolved and they propose an effective technique towards the solution. The meta reviewer recommends acceptance based on the following observations.

The submission proposes a GAN architecture for image inpainting using co-modulation, which is similar to the weight modulation in StyleGAN2 but is conditioned on both the input image and the stochastic variable instead of only the stochastic variable. The main novelty of co-modulation appears to be interesting as well as being generalisable to different tasks. The approach is shown to perform well in the image painting with large-scale missing pixels and some image-to-image translation tasks. Furthermore a new metric P-IDS/U-IDS is proposed to evaluate the perceptual fidelity of inpainted images.